Extending a land-surface model with Sphagnum moss to simulate responses of a northern temperate bog to whole-ecosystem warming and elevated $CO_2$

Xiaoying Shi[1][*], Daniel M. Ricciuto[1], Peter E. Thornton[1], Xiaofeng Xu[2], Fengming Yuan[1], Richard J. Norby[1], Anthony P. Walker[1], Jeffrey Warren[1], Jiafu Mao[1], Paul J. Hanson[1], Lin Meng[3], David Weston[1], Natalie A. Griffiths[1]

[1] Climate Change Science Institute and Environmental Sciences Division, Oak Ridge National Laboratory, Oak Ridge, TN 37831, USA

[2] Biology Department San Diego State University, San Diego, CA, 92182-4614, USA

[3] Department of Geological and Atmospheric Sciences, Iowa State University, Ames, IA, 50011

[*] To whom correspondence should be addressed

Corresponding author's email: shix@ornl.gov

Fax: 865-574-2232

**Abstract**
Mosses need to be incorporated into Earth system models to better simulate

peatland functional dynamics under changing environment. *Sphagnum* mosses are strong
determinants of nutrient, carbon and water cycling in peatland ecosystems. However,
most land surface models do not include *Sphagnum* or other mosses as represented plant
functional types (PFTs), thereby limiting predictive assessment of peatland responses to
environmental change. In this study, we introduce a moss PFT into the land model
component (ELM) of the Energy Exascale Earth System Model (E3SM), by developing
water content dynamics and non-vascular photosynthetic processes for moss. The model
was parameterized and independently evaluated against observations from an
ombrotrophic forested bog as part of the Spruce and Peatland Responses Under Changing
Environments (SPRUCE) project. Inclusion of a *Sphagnum* PFT with some *Sphagnum*
specific processes in ELM allows it to capture the observed seasonal dynamics of
*Sphagnum* gross primary production (GPP), albeit with an underestimate of peak GPP.
The model simulated a reasonable annual net primary production (NPP) for moss but
with less interannual variation than observed, and reproduced above ground biomass for
tree PFTs and stem biomass for shrubs. Different species showed highly variable
warming responses under both ambient and elevated atmospheric $CO_2$ concentrations,
and elevated $CO_2$ altered the warming response direction for the peatland ecosystem.
Microtopography is critical: *Sphagnum* mosses on hummocks and hollows were
simulated to show opposite warming responses (NPP decreasing with warming on
hummocks, but increasing in hollows), and hummock *Sphagnum* was modeled to have
strong dependence on water table height. Inclusion of this new moss PFT in global ELM
simulations may provide a useful foundation for the investigation of northern peatland
carbon exchange, enhancing the predictive capacity of carbon dynamics across the
regional and global scales.
**Copyright statement**
**1. Introduction**
Boreal peatlands store at least 500 Pg of soil carbon due to incomplete
decomposition of plant litter inputs resulting from a combination of low temperature and
water-saturated soils. Because of this capacity to store carbon, boreal peatlands have
played a critical role in regulating the global climate since the onset of the Holocene
(Frolking and Roulet, 2007; Yu et al., 2010). The total carbon stock is large but
uncertain: a new estimation of northern peatlands carbon stock of 1055 Pg was recently
reported by Nichols and Peteet (2019). The rapidly changing climate at high latitudes is
likely to impact both primary production and decomposition rates in peatlands,
contributing to uncertainty in whether peatlands will continue their function as net carbon
sinks in the long term (Moore et al., 1998; Turetsky et al., 2002; Wu and Roulet, 2014).
Manipulative experiments and process-based models are thus needed to make defensible
projections of net carbon balance of northern peatlands under anticipated global warming
(Hanson et al, 2017; Shi et al., 2015).

Peatlands are characterized by a ground layer of bryophytes, and the raised or

ombrotrophic bogs of the boreal zone are generally dominated by *Sphagnum* mosses that
contribute significantly to total ecosystem $CO_2$ flux (Oechel and Van Cleve, 1986;
Williams and Flanagan, 1998; Robroek et al., 2009; Vitt, 2014). *Sphagnum* mosses also
strongly affect the hydrological and hydrochemical conditions at the raised bog surface
(Van, 1995; Van der Schaaf, 2002). As a result, microclimate and *Sphagnum* species
interactions influence the variability of both carbon accumulation rates and water and
exchanges within peatland and between peatland and atmosphere (Heijmans et al., 2004a,
2004b; Rosenzweig et al., 2008; Brown et al., 2010; Petrone et al., 2011; Goetz and Price,
2015). Functioning as keystone species of boreal peatlands, *Sphagnum* mosses strongly
influence the nutrient, carbon and water cycles of peatland ecosystems (Nilsson and
Wardle, 2005; Cornelissen et al., 2007; Lindo and Gonzalez, 2010; Turetsky et al., 2010;
Turetsky et al., 2012), and exert a substantial impact on ecosystem net carbon balance
(Clymo and Hayward; 1982; Gorham, 1991; Wieder, 2006; Weston et el., 2015; Walker
et al., 2017; Griffiths et al., 2018).

Numerical models are useful tools to identify knowledge gaps, examine long-term

dynamics, and predict future changes. Earth system models (ESMs) simulate global
processes, including the carbon cycle, and are primarily used to make future climate
projections. Poor model representation of carbon processes in peatlands is identified as a

**Deleted:** water exchanges between and within peatlands

deficiency causing biases in simulated soil organic mass and heterotrophic respiratory
fluxes for current ESMs (Todd-Brown et al., 2013; Tian et al., 2015).  Although most
ESMs do not include moss, a number of offline dynamic vegetation models and
ecosystem models do include one or more moss plant functional types (PFTs) (Pastor et
al., 2002; Nungesser, 2003; Zhuang et al., 2006; Bond-Lamberty et al., 2007; Heijmans et
al., 2008; Euskirchen et al., 2009; Wania et al., 2009; Frolking et al., 2010). Several
peatland-specific models contain moss species and have been applied globally or at
selected peatland sites. For example, the McGill Wetland Model (MWM) was evaluated
using the measurements at Degerö Stormyr and the Mer Bleue bogs (St-Hilaire et al.,
2010). The peatland version of the General Ecosystem Simulator - Model of Raw Humus,
Moder and Mull (GUESS-ROMUL) was used to simulate the changes of daily $CO_2$
exchange rates with water table position at a fen (Yurova et al., 2007). The PEATBOG
model was implemented to characterize peatland carbon and nitrogen cycles in the Mer
Bleue bog, including moss PFTs but without accounting for microtopography (Wu et al.,
2013a). The CLASS-CTEM model (the coupled Canadian Land Surface Scheme and the
Canadian Terrestrial Ecosystem Model), which includes a moss layer as the first soil
layer, was applied to simulate water, energy and carbon fluxes at eight different peatland
sites (Wu et al., 2016). The IAP-RAS (Institute of Applied Physics – Russian Academy
of Sciences) wetland methane ($CH_4$) model with a 10 cm thick moss layer (Mokhov et al.
2007) was run globally to simulate the distribution of $CH_4$ fluxes (Wania et al., 2013).
The CHANGE model (a coupled hydrological and biogeochemical process simulator),
which includes a moss cover layer (Launiainen et al., 2015), was used to investigate the
effect of moss on soil temperature and carbon flux at a tundra site in Northeastern Siberia

(Park et al., 2018). Chadburn et al. (2015) added a surface layer of moss to JULES land surface model to consider the insulating effects and treated the thermal conductivity of moss depending on its water content to investigate the permafrost dynamics. Porada et al. (2016) integrated a stand-alone dynamic non-vascular vegetation model LiBry (Porada et al., 2013) to land surface scheme JSBACH, but JSBACH mainly represent bryophyte and lichen cover on upland forest, not for peatland ecosystem without including an organic soil layer. Druel et al. (2017) investigated the vegetation-climate feedbacks in high latitudes by introducing a non-vascular plant type representing mosses and lichens to the global land surface model ORCHIFEE. Moreover, those models did not consider microtopography and the lateral transports between hummocks and hollows. Two models, the "ecosys" model (Grant et al., 2012) and CLM_SPRUCE (Shi et al., 2015), have been parameterized to represent peatland microtopographic variability (e.g., the hummock and hollow microterrain characteristic of raised bogs) with lateral connections across the topography. Prediction of water table dynamics in the "ecosys" model is constrained by specifying a regional water table at a fixed height and a fixed distance from the site of interest, thereby missing key controlling factors of a precipitation-driven dynamic water table (Shi et al., 2015). The CLM_SPRUCE model (Shi et al., 2015) was developed to parameterize the hydrological dynamics of lateral transport for microtopography of hummocks and hollows in the raised bog environment of the SPRUCE (Spruce and Peatland Responses Under Changing Environments) experiment (Hanson et al., 2017). That model version did not include the biophysical dynamics of *Sphagnum* moss, and used a prescribed leaf area instead of allowing leaf area to evolve prognostically.

In this study, we introduce a new *Sphagnum* moss PFT into the model, and migrate
the entire raised-bog capability into the new Energy Exascale Earth System Model
(E3SM), specifically into version 1 of the E3SM land model (ELM v1, Ricciuto et al.,
2018). The objectives of this study are to: 1) introduce a *Sphagnum* PFT to the ELM
model with additional *Sphagnum*-specific processes to better capture the peatland
ecosystem; and 2) apply the updated ELM to explore how an ombrotrophic, raised-dome
bog peatland ecosystem will respond to different scenarios of warming and elevated
atmospheric $CO_2$ concentration.
**2. Model description**
**2.1 Model provenance**
ELM v1 is the land component of E3SM v1, which is supported by the US
Department of Energy (DOE). Developed by multiple DOE laboratories, E3SM consists
of atmosphere, land, ocean, sea ice, and land ice components, linked through a coupler
that facilitates across-component communication (Golaz et al., 2019). ELM was
originally branched from the Community Land Model (CLM4.5, Oleson et al., 2013),
with new developments that include  representation of coupled carbon, nitrogen, and
phosphorus controls on soil and vegetation processes, and new plant carbon and nutrient
storage pools (Ricciuto et al., 2018; Yang et al., 2019; Burrows et al., 2020). Inputs of
new mineral nitrogen of ELM are from atmospheric deposition and biological nitrogen
fixation. The fixation of new reactive nitrogen from atmospheric $N_2$ by soil
microorganisms is an important component of nitrogen budgets. ELM follows the
approach of Cleveland et al. (1999) that uses an empirical relationship of biological

nitrogen fixation as a function of net primary production to predict the nitrogen fixation.

The model version used in this study is designated ELM SPRUCE, and includes the new implementation of Sphagnum mosses as well as the hydrological dynamics of lateral transport between hummock and hollow microtopographies. The implementation has been parameterized based on observations from the S1-Bog in northern Minnesota, USA, as described by Shi et al. (2015), with additional details provided below.

2.2 Non-vascular plants: *Sphagnum* mosses

To represent non-vascular plant the *Sphagnum* mosses, we modified the C3 artic grasses equations as follows. We considered *Sphagnum* biomass to be represented mainly by leaf and stem carbon (only a very shallow root). In addition, we modified the vascular C3 arctic grasses equations for photosynthesis and stomatal conductance (see the below new model development), and the associated parameters as reported by Table 1-3. We use the same framework as for C3 artic grasses, but the Ball-Berry slope term is assumed to be zero and the intercept term is the conductance term as a function of water content of *Sphagnum* mosses. For all other processes like the evapo(transpi)ration and associated parameters not described below, we used the C3 artic grasses equations (reported by Oleson et al., 2013). Drying impacts the conductance and affects evapo(transpi)ration of the internal water. The SLA and leaf C:N ratio parameters are strong controls on Vcmax, and therefore overall productivity and Sphagnum moss LAI. The high sensitivities occur because LAI is a strong control on evapo(transp)iration.

**2.3 New model developments**

**2.3.1 Water content dynamics of *Sphagnum* mosses**

The main sources for water content of *Sphagnum* mosses are passive capillary

water uptake from peat, and interception of atmospheric water on the capitulum (growing
tip of the moss) (Robroek et al. 2007). Capillary water uptake, the internal *Sphagnum*
moss water content, is modeled as functions of soil water content and evaporation losses.
Water intercepted on the *Sphagnum* moss capitulum is modeled as a function of moss
foliar biomass, current canopy water, water drip, and evaporation losses.

Since evaporation at the *Sphagnum* surface depends on atmospheric water vapor

deficit, moss-atmosphere conductance and available water pool which depends on
capillary wicking of water up to the surface, At SPRUCE, the peat volumetric water
content is measured at several depths using automated sensors (model 10HS, Decagon
Devices, Inc., Pullman, WA) calibrated for the site-specific upper peat soil using
mesocosms (reference Figure S1, Hanson et al. 2017). During those calibrations, we
periodically sampled the surface *Sphagnum* for gravimetric water content and water
potential using a dew point potentiometer (WP4, Decagon Devices, Inc.), which also
provided a surface soil water retention curve. The destructive sampling of surface
*Sphagnum* was primarily hummock species but did included some hollow species. The
automated measurements of peat water content at 10 cm depth were shown to be a good
indicator of surface *Sphagnum* water content (Fig. 1).  Based on this relationship, we
model the water content of *Sphagnum* moss due to capillary rise ($W_{internal}$) (g water /g
dry moss) as:
$W_{internal} = 0.3933 + 7.6227/(1 + \exp(-(Soil_{vol} - 0.1571))/0.018$         (1)
where $Soil_{vol}$ is the averaged volumetric soil water of modeled soil layers nearest the
10cm depth horizon (layers 3 and 4 in the ELM v1 vertical layering scheme).

**Formatted:** Font: Italic

**Deleted:** Since evaporation at the *Sphagnum* surface depends on capillary wicking of water up to the surface and atmospheric water vapor deficit, we developed a relationship between measured soil water content at depth, and surface *Sphagnum* water content

**Deleted:** W

**Deleted:** .

The *Sphagnum* moss surface water ($W_{surface}$) was calculated using the model

predicted canopy water and the dry foliar biomass as:

$$W_{surface} = can\_water/fmass \qquad (2)$$

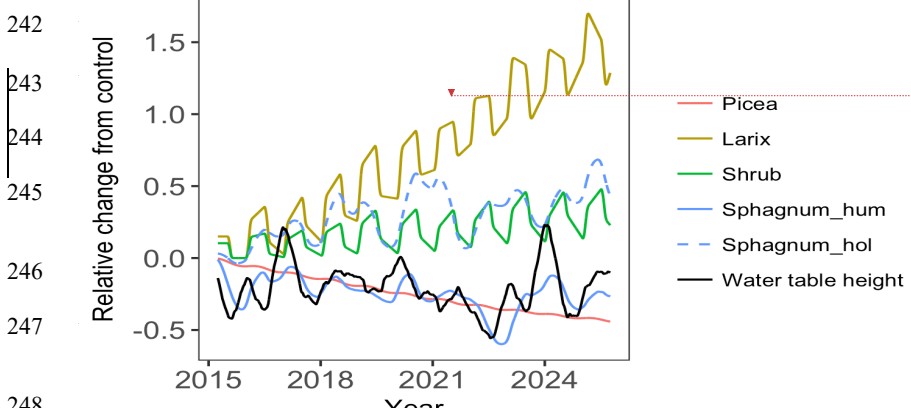

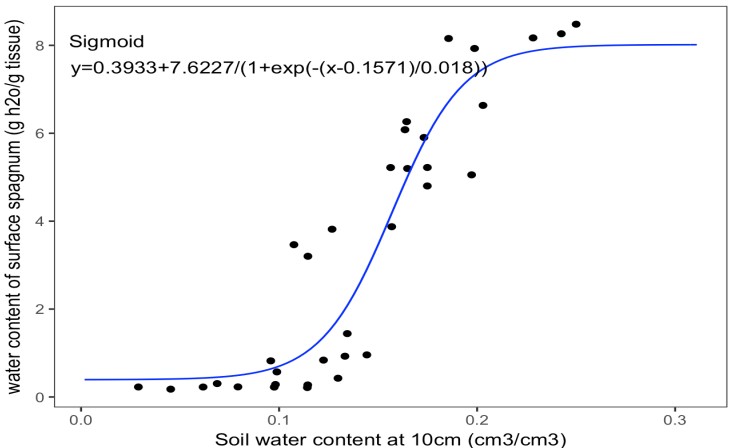

Figure 1. The measured relationship between soil water content at depth and the water content of
surface Sphagnum based on destructive sampling.

**2.3.2 Modeling *Sphagnum* CO₂ conductance and photosynthesis**

Deleted: 2

ELM_SPRUCE computes photosynthetic carbon uptake (gross primary production, or GPP) for each vascular PFT on a half-hourly time step, based on the Farquhar biochemical approach (Farquhar et al., 1980; Collatz et al., 1991, 1992), with implementation as described by Oleson et al. (2013). While, *Sphagnum* lacks a leaf cuticle and stomata that regulate water loss and $CO_2$ uptake in vascular plants (Titus et al. 1983). The primary transport pathway for $CO_2$ is through the cells and is analogous to mesophyll conductance in higher plants. Thus, we calculate the total conductance to $CO_2$ for *Sphagnum* mosses by using total water content following the method reported by Williams and Flanagan (1998) described as below. Goetz and Price (2015) also indicated that capillary rise through the peat is essential to maintain a water content sufficient for photosynthesis for *Sphagnum* moss species, but that atmospheric inputs can provide small but critical amounts of water for physiological processes.

Formatted: Font: Italic

Formatted: Not Superscript/ Subscript

Deleted: The internal water content of *Sphagnum* mosses is observed to affect photosynthesis by constraining the length of the diffusive path for $CO_2$ through the variably-hydrated external hyaline cells to the carbon fixation sites (Robroek et al., 2009; Rydin and Jeglum, 2006). Goetz and Price (2015) also indicated that capillary rise through the peat is essential to maintain a water content sufficient for photosynthesis for *Sphagnum* moss species, but that atmospheric inputs can provide small but critical amounts of water for physiological processes. *Sphagnum* lacks a leaf cuticle and stomata that regulate water loss and $CO_2$ uptake in vascular plants (Titus et al. 1983). The primary transport pathway for $CO_2$ is through the cells and is analogous to mesophyll conductance in higher plants.

The stomatal conductance for vascular plant types in ELM_SPRUCE is derived from the Ball-Berry conductance model (Collatz et al., 1991). That model relates stomatal conductance to net leaf photosynthesis, scaled by the relative humidity and the $CO_2$ concentration at the leaf surface. The stomatal conductance ($g_s$) and boundary layer conductance ($g_b$) are required to obtain the internal leaf $CO_2$ partial pressure ($C_i$) of vascular PFTs:

$$C_i = C_a - \left(\frac{1.4 g_s + 1.6 g_b}{g_s\, g_b}\right) P_{atm} A_n \tag{4}$$

where $C_i$ is the internal leaf $CO_2$ partial pressure, $C_a$ is the atmospheric $CO_2$ partial

pressure, $A_n$ is leaf net photosynthesis ($\mu$ mol $CO_2$ m$^{-2}$ s$^{-1}$) $P_{atm}$ is the atmospheric
pressure, $g_s$ is the leaf stomatal conductance, $g_b$ is the leaf boundary layer conductance,
and values 1.4 and 1.6 are the ratios of the diffusivity of $CO_2$ to $H_2O$ for stomatal
conductance  and the leaf boundary layer conductance, respectively.

For *Sphagnum* moss photosynthesis, we followed the method from the McGill

Wetland Model (St-Hilaire et al. 2010; Wu et al., 2013), which is based on the effects of
*Sphagnum* moss water content on photosynthetic capacity (Tenhunen et al., 1976) and
total conductance of $CO_2$ (Williams and Flanagan, 1998), and replaces the stomatal
conductance representation used for vascular PFTs.
$$C_i = C_a - \frac{P_{atm}A_n}{g_{tc}} \qquad\qquad (5)$$
The total conductance to $CO_2$ ($g_{tc}$) was determined from a least-squares regression
described by Williams and Flanagan (1998) as:
$$g_{tc} = -0.195 + 0.134W_{total} - 0.0256W_{total}{}^2 + 0.0028W_{total}{}^3 -$$
$$0.0000984W_{total}{}^4 + 0.00000168W_{total}{}^5 \qquad\qquad (6)$$
where $W_{total}$ is as defined in equation (3). This relationship is only valid up to the
maximum water holding capacity of mosses. To be noted that we assume that the
boundary layer conductance is greater than moss surface layer conductance, and the moss
surface layer conductance is greater than chloroplast conductance.

In addition to the water content, the effects of moss submergence were taken into

account in the calculation of moss photosynthesis. Walker et al. (2017) reported
significant impacts of submergence on measured *Sphagnum* GPP and modeled the effect
by modifying the *Sphagnum* leaf (stem) area index. Submergence in Walker et al. (2017) was
expressed as photosynthesising stem area index (SAI) as a logistic function of water table depth.
A maximum SAI of 3 was used and the parameter combination that most closely described the
GPP data gave a range of water table depth from -10 cm for complete submergence and SAI of
~2.5 at 10 cm. This allowed for a range of processes such as floatation of *Sphagnum* with the
water table, and adhesion of water to the *Sphagnum* capitula. For simplicity, in
ELM_SPRUCE, we calculated such impacts on *Sphagnum* GPP directly as a function of
the height of simulated surface water, assuming that GPP from the submerged portion of
photosynthetic tissue is negligible. GPP is thus reduced linearly according to the
following equation:
$GPP_{sub} = GPP_{orig} * (h_{moss} - H_2O_{sfc})$                             (7)
where $GPP_{sub}$ is the GPP corrected for submergence effects, $GPP_{orig}$ is the original GPP,
$H_2O_{sfc}$ is the surface water height, and $h_{moss}$ is the height of the photosynthesizing
*Sphagnum* layer above the soil surface, set to 5cm in our simulations. If $H_2O_{sfc}$ is equal to
or greater than $h_{moss}$, GPP is reduced to zero. Because in our simulations surface water is
never predicted to occur in the hummocks, in practice this submergence effect only
affects the moss GPP in the hollows.
**3. Methods**
**3.1 Site Description**

We focused on a high C, ombrotrophic peatland (the S1-Bog) that has a perched

water table with limited groundwater influence (Sebestyen et al. 2011, Griffiths and
Sebestyen, 2016). This southern boreal bog is located on the Marcell Experimental

Formatted: Font: Italic

Formatted: Font: Italic

Forest, approximately 40 km north of Grand Rapids, Minnesota, USA (47.50283 degrees
latitude, -93.48283 degrees longitude) (Sebestyen et al. 2011), and is the site of the
SPRUCE climate change experiment (http://mnspruce.ornl.gov; Hanson et al., 2017). The
S1-Bog has a raised hummock and sunken hollow microtopography, and it is nearly
covered by *Sphagnum* mosses. *S. angustifolium* (C.E.O. Jensen ex Russow) and *S. fallax*
(Klinggr.) occupy 68% of the moss layer and exist in both hummocks and hollows.  *S.*
*magellenicum* (Brid.) occupies ~20% of the moss layer and is primarily limited to the
hummocks (Norby et al., 2019). The vascular plant community at the S1-Bog is
dominated by the evergreen tree *Picea mariana* (Mill.) B.S.P, the deciduous tree *Larix*
*laricina* (Du Roi) K. Koch, and a variety of ericaceous shrubs. Trees are present due to
natural regeneration following strip cut harvesting in 1969 and 1974 (Sebestyen et al.,
2011). The soil of this peat bog is the Greenwood series, a Typic Haplohemist
(https://websoilsurvey.sc.egov.usda.gov), and its average peat depth is 2 to 3 m
(Parsekian et al., 2012)

Northern Minnesota has a subhumid continental climate with average annual

precipitation of 768 mm and annual air temperature of 3.3 °C for the time period from
1965 to 2005. Mean annual air temperatures at the bog have increased about 0.4 °C per
decade over the last 40 years (Verry et al., 2011).
**3.2 Field measurements**

Multiple observational pre-treatment data (the data were collected prior to

initiation of the warming and $CO_2$ treatments) were used in this study. Flux-partitioned
GPP of *Sphagnum* mosses was derived from measured hourly *Sphagnum*-peat net
ecosystem exchange (NEE) flux (Walker et al., 2017). The GPP – NEE relationship was

Formatted: Subscript

also evaluated using observed vegetation growth and productivity allometric and biomass
data on tree species, stem biomass for shrub species (Hanson et al., 2018a and b), and
*Sphagnum* pre-treatment net primary productivity (NPP) (Norby et al., 2019).
ELM_SPRUCE was driven by climate data (temperature, precipitation, relative humidity,
solar radiation, wind speed, pressure and long wave radiation) from 2011 to 2017
measured at the SPRUCE S1-Bog (Hanson et al., 2015a and b). The surface weather
station is outside of the enclosures and not impacted by the experimental warming
treatments that began in 2015. These data are available at https://mnspruce.ornl.gov/.
**3.3 Simulation of the SPRUCE experiment**

Based on measurements at the SPRUCE site, ELM_SPRUCE includes four

PFTs: boreal evergreen needleleaf tree (*Picea)*, boreal deciduous needleleaf tree (*Larix)*,
boreal deciduous shrub (representing several shrub species), and the newly introduced
*Sphagnum* moss PFT. Currently ELM_SPRUCE does not include light competition
among multiple PFTs, and thus does not represent cross-PFT shading effects. Our model
also allows the canopy density of PFTs to change prognostically, and their fractional
coverage is held constant. We used measurements from *Sphagnum* moss collected at a
tussock tundra site in Alaska (Hobbie 1996) to set several of the model leaf litter
parameters for our simulations (Table 1). The values for other parameters have been
optimized based on observations at the SPRUCE site (Table 2 and 3, optimization
methods described in section 3.4). We prescribe both hummock and hollow
https://scratch.mit.edu/projects/411435898microtopographies to have the same fractional
PFT distribution. Consistent with Shi et al. (2015), hummocks and hollows were
modeled on separate columns with lateral flow of water between them. All the
ELM_SPRUCE simulations were conducted using a prognostic scheme for canopy
phenology (Olesen et al., 2013).
The SPRUCE experiment at the S1-Bog consists of combined manipulations of
temperature (various differentials up to +9 °C above ambient) and atmospheric $CO_2$
concentration (ambient and ambient + 500 ppm) applied in 12 m diameter x 8 m tall
enclosures constructed in the S1-Bog. The whole-ecosystem warming began in August
2015, elevated $CO_2$ started from June 2016, and various treatments are envisioned to
continue until 2025. Extensive pre-treatment observations at the site began in 2009.
For the ELM_SPRUCE, we continuously cycled the 2011-2017 climate forcing
(see section 3.2) to equilibrate carbon and nitrogen pools under pre-industrial
atmospheric $CO_2$ concentrations and nitrogen deposition, and then launched a simulation
starting from year 1850 through year 2017. This transient simulation includes historically
varying $CO_2$ concentrations, nitrogen deposition, and the land-use effects of a strip cut
and harvest at the site in 1974. These simulations were used to compare model
performance with pre-treatment observations.  A subset of these observations was also
used for optimization and calibration (section 3.4).
To investigate how the bog vegetation may respond to different warming
scenarios and elevated atmospheric $CO_2$ concentrations, we performed 11 model runs
from the same starting point in year 2015. These simulations were designed to reflect the
warming treatments and $CO_2$ concentrations being implemented in the SPRUCE
experiment enclosures. The model simulations include one ambient case (both ambient
temperature and $CO_2$ concentration), and five simulations with modified input air
temperatures to represent the whole-ecosystem warming treatments at five levels (+0 ºC,
+2.25 ºC, +4.50 ºC, +6.75 ºC and +9.00 ºC above ambient) and at ambient $CO_2$, and
another five simulations with the same increasing temperature levels and at elevated $CO_2$
(900 ppm). In the treatment simulations, we also considered the passive enclosure
effects, which reduce incoming shortwave and increase incoming longwave radiation
(Hanson et al., 2017). Following the SPRUCE experimental design, there was no water
vapor added so that the simulations used constant specific humidity instead of constant
relative humidity across the warming levels. All the treatment simulations were
performed through the year 2025 by continuing to cycle the 2011-2017 meteorological
inputs (with modified temperature and radiation to reflect the treatments) to simulate
future years.
Table 1: Physiological parameters of *Sphagnum* mosses as given in Hobbie 1996

| Parameters | Description | Values |
|---|---|---|
| lflitcn | Leaf litter C:N ratio (gC/gN) | 66 |
| lf_fcel | Leaf litter fraction of cellulose | 0.737 |
| lf_flab | Leaf litter fraction of labile | 0.227 |
| lf_flig | Leaf litter fraction of lignin | 0.036 |


## 3.4. Model sensitivity analysis and calibration

The vegetation physiology parameters in ELM_SPRUCE were originally derived
from CLM4.5 and its predecessor, Biome-BGC, and represent broad aggregations of
plant traits over many species and varied environmental conditions (White et al., 2000).
To achieve reasonable model performance at SPRUCE, site-specific parameters and
targeted parameter calibration are needed. Since the ELM_SPRUCE contains over 100
uncertain parameters, parameter optimization is not computationally feasible without first
performing some dimensionality reduction. Based on previous ELM sensitivity analyses
(e.g., Lu et al., 2018; Ricciuto et al., 2018; Griffiths et al., 2018), we chose 35 model
parameters for further calibration (Tables 2 and 3). An ensemble of 3000 ELM_SPRUCE
simulations were conducted using the procedure described in 3.3, with each ensemble
member using a randomly selected set of parameter values within uniform prior ranges.
This model ensemble was first used to construct a polynomial chaos surrogate model,
which was then used to perform a global sensitivity analysis (Sargsyan et al., 2014;
Ricciuto et al., 2018). Main sensitivity indices, reflecting the proportion of output
variance that occurs for each parameter, are described in section 4.1.
To minimize potential biases in model predictions of treatment responses, we
calibrated the same 35 model parameters using pre-treatment observations as data
constraints. We employed a quantum particle swarm optimization (QPSO) algorithm (Lu
et al., 2018). While this method does not allow for the calculation of posterior prediction
uncertainties, it is much more computationally efficient than other methods such as
Markov Chain Monte Carlo. The constraining data included year 2012-2013 tree growth
and biomass (Hanson et al. 2018a), year 2012-2013 shrub growth and biomass (Hanson
et al., 2018b), year 2012 and 2014 *Sphagnum* net primary productivity (Norby et al.,
2017, 2019), enclosure-averaged leaf area index by PFT (year 2011 for tree and year
2012 for shrub and *Sphagnum*), and year 2011-2013 water table depth (WTD)
observations, aggregated to seasonal averages (Hanson et al., 2015b). The goal of the
optimization is to minimize a cost function, which we define here as a sum of squared
errors over all observation types weighted by observation uncertainties. When
observation uncertainties were not available, we assumed a range of ±25% from the
default value. Site measurements were also used to constrain the ranges of two
parameters: *leafcn* (leaf carbon to nitrogen ratio) and *slatop* (specific leaf area at canopy
top). The uniform prior ranges for these parameters represent the range of plot to plot
variability. Optimized parameter values are shown in Table 2 and 3. Section 4 reports the
results of simulations using these optimized parameters, which were used to perform a
spinup, transient (1850-2017) and set of 11 treatment simulations (2015-2025) as
described above.
Table 2: PFT-specific optimized model parameters

| Parameter | Description | *Sphagnum* | *Picea* | *Larix* | Shrub | Range |
|---|---|---|---|---|---|---|
| flnr | Rubisco-N fraction of leaf N | 0.2906 | 0.0678 | 0.2349 | 0.2123 | [0.05,0.30] |
| croot_stem | Coarse root to stem allocation ratio | N/A | 0.2540 | 0.1529 | 0.7540 | [0.05,0.8] |
| stem_leaf[1] | Stem to leaf allocation ratio | N/A | 1.047 | 1.016 | 0.754 | [0.3,2.2] |
| leaf_long | Leaf longevity (yr) | 0.9744 | 5[3] | N/A | N/A | [0.75, 2.0] |
| slatop | Specific leaf area at canopy top ($m^2$ $gC^{-1}$) | 0.00781 | 0.00462 | 0.0128 | 0.0126 | [0.004,0.04] |
| leafcn | Leaf C to N ratio | 35.56 | 70.17 | 64.84 | 33.14 | [20,75] |
| froot_leaf[2] | Fine root to leaf allocation ratio | 0.3944 | 0.8567 | 0.3211 | 0.6862 | [0.15, 2.0] |
| mp | Ball-Berry stomatal conductance slope | N/A | 7.50 | 9.32 | 10.8 | [4.5, 12] |

Optimized values of PFT-specific parameters. The range column values in brackets indicate the range of
acceptable parameter values used in the sensitivity analysis and the optimization across all four PFTs in the
format [minimum, maximum]. N/A indicates that parameter is not relevant for that PFT.
[1] for tree PFTs, this parameter depends on NPP. The value shown is the allocation at an NPP of 800 gC $m^{-2}$
$yr^{-1}$.
[2] the fine root pool is used as a surrogate for non-photosynthetic tissue in *Sphagnum*
[3] This parameter was not optimized; we used the default value.


Table 3: Non PFT-specific optimized model parameters

| | Description | Optimized value | Default | Range |
|---|---|---|---|---|
| r_mort | Vegetation mortality | 0.0497 | 0.02 | [0.005, 0.1] |
| decomp_depth_efolding | Depth-dependence e-folding depth for decomposition (m) | 0.3899 | 0.5 | [0.2, 0.7] |
| $q_{drai,0}$ | Maximum subsurface drainage rate (kg m$^{-2}$ s$^{-1}$) | 3.896e-6 | 9.2e-6[*] | [0, 1e-3] |
| $Q_{10}$_mr | Temperature sensitivity of maintenance respiration | 2.212 | 1.5 | [1.2, 3.0] |
| br_mr | Base rate for maintenance respiration (gC gN m$^2$ s$^{-1}$) | 4.110e-6 | 2.52e-6 | [1e-6, 5e-6] |
| crit_onset_gdd | Critical growing degree days for leaf onset | 99.43 | 200 | [20, 500] |
| lw_top_ann | Live wood turnover proportion (yr$^{-1}$) | 0.3517 | 0.7 | [0.2, 0.85] |
| gr_perc | Growth respiration fraction | 0.1652 | 0.3 | [0.12, 0.4] |
| $r_{drai,0}$ | Coefficient for surface water runoff (kg m$^{-4}$ s$^{-1}$) | 6.978e-7 | 8.4e-8[*] | [1e-9, 1e-6] |

Optimized and default values for non PFT-specific parameters. The range column values in brackets
indicate the range of acceptable parameter values used in the sensitivity analysis and the
optimization in the format [minimum, maximum].
* Previously calibrated value from Shi et al (2015)

**4. Results**
**4.1 Model sensitivity analysis**

Main effect (first-order) sensitivities are shown for eight model output quantities of

interest: Total site gross primary productivity (GPP), GPP for the moss PFT only
(GPP_moss), total site net primary productivity (NPP), NPP for the moss PFT only
(NPP_moss), total site vegetation transpiration (QVEGT), evaporation from the moss
surface (QVEG_moss), net ecosystem exchange (NEE) and site total vegetation carbon
(TOTVEGC) (Fig. 2).  Out of 35 parameters investigated, 25 show a sensitivity index of
at least 0.01 for one of the quantities of interest, and these are plotted on figure 2.  In that
figure, sensitivities are stacked in order from highest to lowest for each variable, with the
height of the bar equal to the sensitivity index.  The first order sensitivities sum to at least
0.95 for all variables, indicating that higher order sensitivities (i.e., contributions to the
sensitivity from combinations of two or more parameters) contribute relatively little to
the variance for these quantities of interest.

According to this analysis, the variance in total site GPP is dominated by three

*Picea* parameters:  the fraction of leaf nitrogen in RuBiCO (*flnr_picea)*, leaf carbon to
nitrogen ratio (*leafcn_picea)* and the specific leaf area at canopy top (*slatop_picea)*.  GPP
sensitivity for the moss PFT is dominated by the same three parameters, but for the moss
PFT instead of *Picea* (*flnr_moss*, *leafcn_moss*, and *slatop_moss*).  For NPP, QVEGT and
NEE, the highest sensitivity the maintenance respiration base rate *br_mr*, similar to
earlier results in Griffiths et al. (2017).  The maintenance respiration temperature
sensitivity $Q_{10}$_mr is also a key parameter for NPP and NEE.  The critical onset growing
degree day threshold (*crit_onset_gdd*), which drives deciduous phenology in the spring
for the *Larix* and shrub PFTs, is an important parameter for NPP and NEE.  *flnr_picea* is
important for both NPP and QVEGT.  For NPP_moss and QVEG_moss, *leafcn_moss* is
and the ratio of non-photosynthesizing tissue to photosynthesizing tissue (*npt_moss)* are
sensitive.  For TOTVEGC and NEE, vegetation mortality (*r_mort)* is also a sensitive
parameter.  For the site-level quantities of interest, at least 10 parameters contribute
significantly to the uncertainty, illustrating the complexity of the model and large number
of processes contributing to uncertainty in SPRUCE predictions.  For the moss variables,
there are some cases where significant sensitivities exist for non-moss PFT parameters.
For example, *leafcn_shrub* is the seventh most sensitive parameter for GPP_moss,
indicating that competition between the PFTs for resources may be important.  In this
case, uncertainty about parameters on one PFT may drive uncertainties in the simulated
productivity of other PFTs.

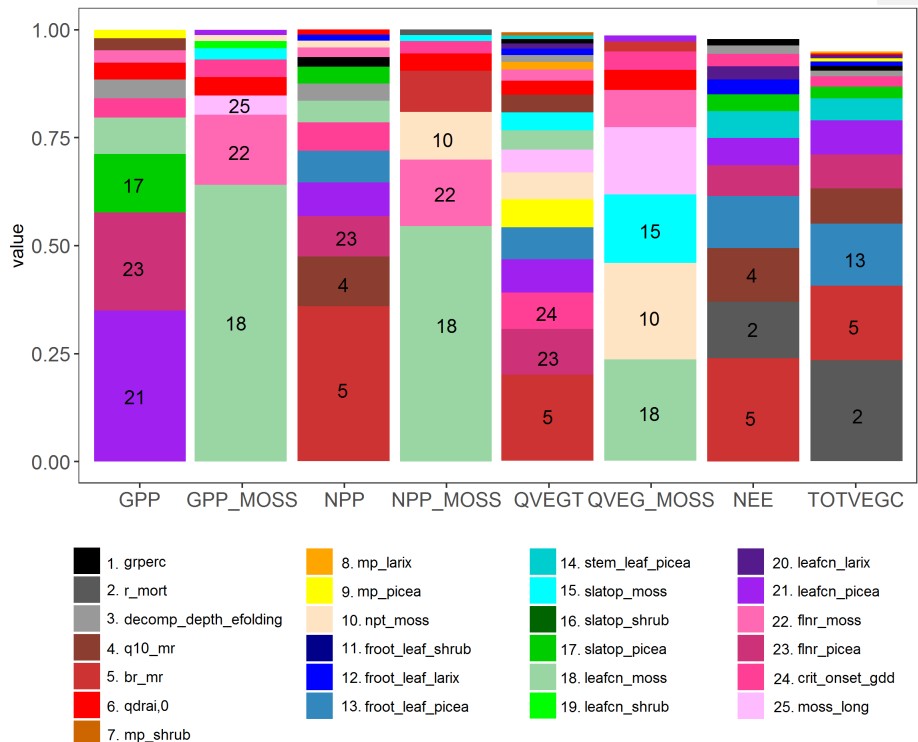


Figure 2 Sensitivity analysis of ELM-SPRUCE for selected parameters (Table 2 and 3).  The
Colored bars indicate the fraction of variance in site gross primary productivity (GPP), moss-only
NPP (GPP_MOSS), site net primary productivity (NPP), moss-only NPP (NPP_MOSS), total
vegetation transpiration (QVEGT), moss evaporation (QVEG_MOSS), site net ecosystem
exchange (NEE) and total vegetation carbon (TOTVEGC) controlled by each parameter.  The
legend shows the top 25 most influential parameters; the remaining parameters not shown have
sensitivities of no more than 0.01 for any of the outputs.  All variables represent 2011-2017
average values over the ambient conditions.  For parameters that are treated as PFT-dependent,
the PFT is indicated with a suffix (picea, larix, shrub or moss)

**4.2 Model evaluation**
Our model simulates GPP for vascular plants and *Sphagnum* moss in both
hummock and hollow settings, with separate calculations for each PFT. Here we use the
model estimate of GPP prior to downregulation by nutrient limitation from the ambient
case, based on recent studies indicating that nutrient limitation effects are occurring
downstream of GPP (Raczka et al. 2016; Metcalfe et al., 2017; Duarte et al. 2017). This
treatment of nutrient limitation on GPP has been modified in a more recent version of
ELM, and our moss modifications will be merged to that version as a next step. For now,
by referring to the pre-downregulation GPP we are capturing the most significant impact
of those changes for the purpose of comparison to observations.
Our model simulated two seasonal maxima of *Sphagnum* moss GPP, one at the
end of May, and the other in August (Figure 3). Both peaks are lower than the maximum
of observed (flux-partitioned) GPP, which occurs in August.  Based on results of the
sensitivity analysis, it could be that the base rate for maintenance respiration for moss is
too high, causing an underestimate of NPP and biomass, which leads to a low bias in
peak GPP.
During June and October, observations suggest that ELM_SPRUCE over-predicts
GPP. The model does limit GPP as a function of the depth of standing water on the bog
surface (Eq. 7). The water table height (WTH) above the bog surface is being predicted
by the model (dashed red line in Fig. 3), and while the seasonal pattern of higher water
table in the spring and lower water table in the fall agrees well with observations (dashed
black line in Fig. 3), the predicted WTH is generally too low by 5-10 cm. The modeled
WTH here is for hollow. We turned off the lateral transport when there is ice on the soil
layers above the water table to avoid an unreasonable amount of ice accumulation on the
frozen layers, which results in there is no flow from hummock to hollow. Forcing the
modeled GPP to respond to observed WTH (during the period with observations) gives a
pattern of increasing GPP through June and July which is more consistent with
observations (blue line in Fig. 3). We do not have observations for GPP earlier than June,
due to limitations of the instrumentation when the bog surface is flooded.

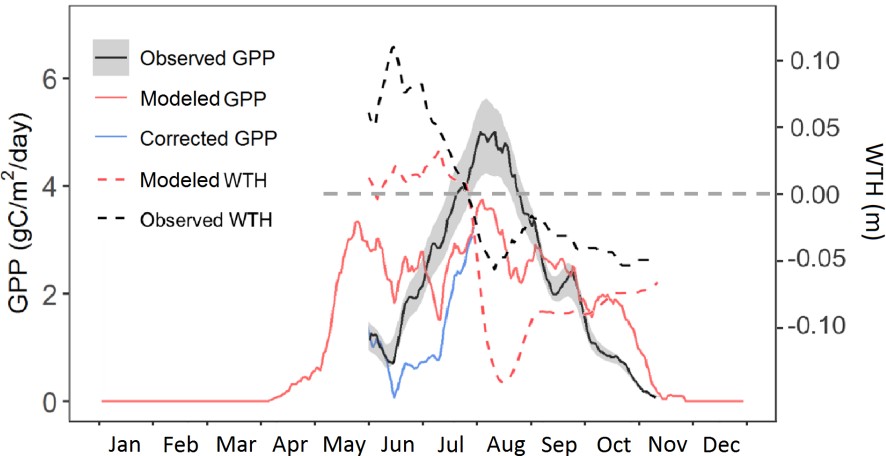

Figure 3. Predicted GPP (red solid line) compared with flux-partitioned GPP (black solid line,
GPP data was not used in the parameters optimization) of *Sphagnum* mosses for the year 2014.
The blue line is the predicted GPP corrected with the observed water table height. The dashed
black and red lines are observed and modeled water table height (the dashed gray line is the
hollow surface).
The model simulated reasonable annual values for *Sphagnum* NPP for the period

2014-2017 but showed much lower NPP compared with observation (139 vs. 288 g
C/m²/yr) for the year 2012 (Fig. 4a). Measurement uncertainties are larger in 2016-2017
than in earlier years, perhaps related to a new measurement protocol for those years, and
the model estimates are within measurement uncertainty bounds for years 2014-2017
(Griffiths et al., 2018; Norby et al., 2019). The observed *Sphagnum* NPP was measured at
different plots and each plot included different species abundances. As a result, the scaled
NPP includes spatial variations and uncertainty in species distribution (Norby and Childs,

2017).

Simulated tree above ground biomass is within the observed inter-plot variability

(Fig. 4b). Observations suggest an increasing trend in tree biomass, which was not
predicted by the model. The optimized parameters show increased mortality and
autotrophic respiration rate parameters compared to the default model (Table 3), which
causes the simulations to approach steady state relatively quickly after the 1974
disturbance.  However, the sensitivity analysis also identifies theses mortality and
maintenance respiration parameters as highly sensitive, therefore this simulated response
is uncertain.  For the shrub stem carbon, the simulated mean from year 2012 to 2015 was
140.4 g C/m², slightly higher than the observation (133.9 g C/m²) but well within the
observed range of inter-plot variability (Fig. 4c).

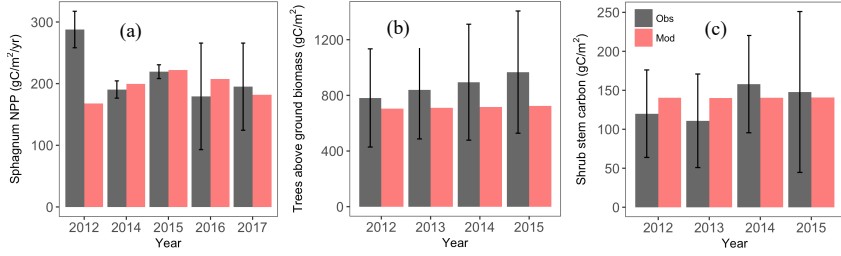


Figure 4. Predicted (red bars) *Sphagnum* NPP (left), aboveground tree biomass (middle) and shrub stem carbon (right) compared with the observations (black bars). Observed NPP data are based on growth of 12-17 bundles of 10 Sphagnum stems in 2012-2015 (unpublished data) and in two ambient plots by the method described by Norby et al. (2019) in 2016-1017 (data in Norby et al. 2017). The Sphagnum NPP data of year 2015-2017, and aboveground tree biomass and shrub stem carbon of year 2014-2015 are independent of the related parameters opitimizaton.

**4.3 Simulated carbon cycle response to warming and elevated atmospheric $CO_2$**

**concentration**

Different PFTs demonstrated different warming responses for both ambient $CO_2$ and elevated $CO_2$ concentration conditions (Fig. 5). Both *Larix* and shrub NPP increased with warming under both $CO_2$ concentration conditions (Fig. 5 b, c, h and i). In addition, $CO_2$ fertilization stimulates the growth of these two PFTs and the fertilization effect further increases with warming (Fig. S1, GPP increases more under elevated $CO_2$ condition than the ambient case). In contrast, *Picea* NPP decreased with warming levels (Fig. 5 a and g) for both $CO_2$ conditions. For *Sphagnum,* NPP decreased in hummocks but increased in hollows with increasing temperature (Fig. 5 d, e, j and k). The $CO_2$ fertilization also stimulate the grow of the Picea and *Sphagnum* PFTs (Fig. 5 a, d, e g, j and k). The enclosure-total NPP for all PFTs responded differently to the warming only and warming with elevated $CO_2$ (Fig. 5 f and l). The enclosure-total NPP for each warming level changed less under the ambient $CO_2$ condition than those with elevated $CO_2$ condition, and NPP decreased with warming in most of years under ambient $CO_2$ condition but increased under elevated $CO_2$ condition (Fig.5 f and l). This result demonstrated that the elevated $CO_2$ scenario changes the sign of the NPP warming response for the bog peatland ecosystem.

**Formatted:** Font: Italic

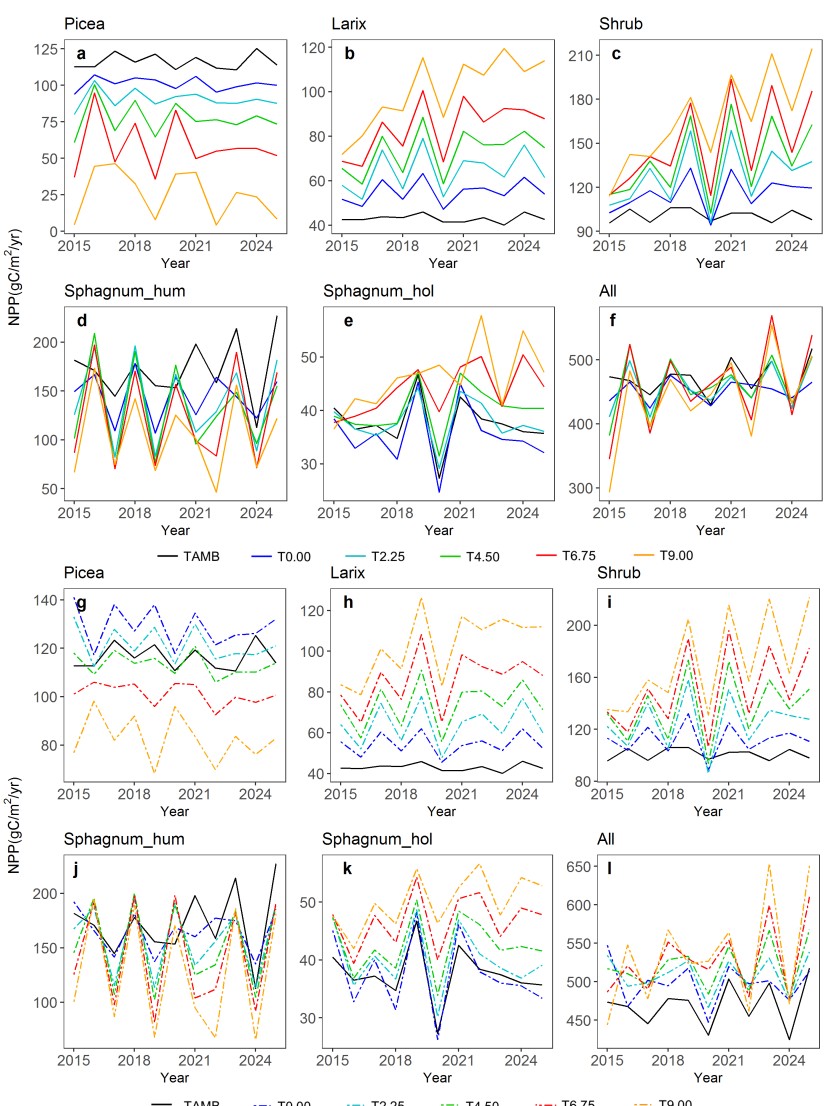


Figure 5 predicted NPP response to warming with ambient atmospheric $CO_2$ (a-f, solid lines) and
warming with elevated atmospheric $CO_2$ concentration (g-l, dash lines), the black solid line
TAMB is the ambient temperature and $CO_2$ case, T0.00 to T9.00 means increasing temperature
from 0°C to 9°C

Compared with the ambient biomass, the biomass of black spruce (*Picea*)

significantly decreased but the biomass of *Larix* significantly increased under the greatest
warming treatment (+9.00°C, Fig.6). Biomass of shrub and hollow *Sphagnum* also
increased, but less than did *Larix*. The hummock *Sphagnum* biomass also showed strong
correlation with water table height at roughly a 3-month lag (the maximum correlation
occurs with an 82-day lag, $R^2$=0.56). NPP is allocated instantaneously into biomass.  A
positive NPP anomaly caused by water table shifts leads to higher LAI, which also
increases future productivity for some amount of time even if the water table returns to
normal.  *Sphagnum* biomass has a 1-year turnover time in the simulation.  This
combination of effects leads to a roughly 3-month timelag. Due to the relative lower
height of the water table in the hummock than the hollow, the simulated hummock
*Sphagnum* were more significantly water-stressed than the hollow *Sphagnum* as the water
table height declines. This is consistent with multiple studies finding an increase in
temperatures associated with drought (low water table height) reducing *Sphagnum*
growth (Bragazza et al., 2016; Granath et al., 2016; Mazziotta et al., 2018).  We plotted
the predicted canopy evaporation for hummock and hollow *Sphagnum* responses to
warming and found that both hummock and hollow *Sphagnum* canopy evaporation
increase with warming for both ambient and elevated atmospheric $CO_2$ conditions despite
the Larix and shrubs are growing with warming. Moreover, the hollow *Sphagnum* canopy
evaporation warming response is stronger than that of the hummock *Sphagnum* (Fig. S2).
In summary, the growth of bog vegetation is predicted to have species-specific warming
responses that differ in sign and magnitude.

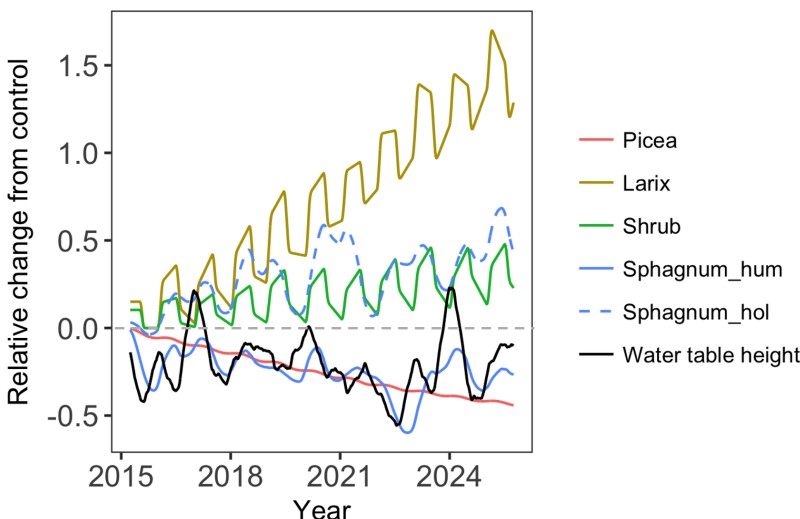

Figure 6 The relative changes of biomass for different PFTs and water table height (the weighted
average between hummock and hollow) between +9.00 ºC treatment case and the ambient case
(+9.00 ºC / ambient – 1)

**5. Discussion**

*Sphagnum* moss is the principal plant involved in the peat accumulation in peatland
ecosystems, and effective characterization of its biophysical and physiological responses
has implications for predicting peatland and global carbon, water and climate feedbacks.
This study moves us closer to our long-term goal of improving the prediction of peatland
water, carbon and nutrient cycles in ELM_SPRUCE, by introducing a new *Sphagnum*
moss PFT, implementing water content dynamics and photosynthetic processes for this
nonvascular plant. The *Sphagnum* model development combined with our previous
hummock-hollow microtopography representation and laterally-coupled two-column
hydrology scheme enhance the capability of ELM_SPRUCE in simulating high-carbon
wetland hydrology and carbon interactions and their responses to plausible environmental
changes.

**5.1 Uncertainties in simulating *Sphagnum* productivity**

Our predicted peak GPP is similar to the results found by Walker et al. (2017)
when they calculated the internal resistance to $CO_2$ diffusion as a function of *Sphagnum*
water content using a stand-alone photosynthesis model. In both cases, the predicted peak
GPP is lower than observations. Walker et al. (2017) were, however, able to capture the
observed peak magnitude with a combination of light extinction coefficient, canopy
clumping coefficient, maximum shoot area index (SAI), and a logistic function
describing the effective *Sphagnum* SAI in relation to water table. Here we used model
default values for the light extinction and canopy clumping coefficients.  While the water
table impacts *Sphagnum* productivity in our simulation, modeled leaf (or shoot) area
index (LAI) is mainly controlled by NPP and turnover. In addition, we use the default
formulation for acclimation of Vcmax in ELM which is based on a 10-day mean growing
temperature.  At this point we don't have sufficient measurements to test this assumption,
but we can prioritize these measurements in the future. *Sphagnum* temperature is
computed from surface energy balance but because the current model doesn't have the
capacity to estimate the shading effects from trees and shrubs, this may be overestimated.
Moreover, biases in predicted water table height contribute to errors in the calculated
submergence effect.  Improving these biases and assuming an exponential rather than a
linear $CO_2$ uptake profile may improve representation of the submergence effect.  All
these aspects may be attribute to the biases of the simulated *Sphagnum* GPP. We can
consider this in the future when we have more detailed measurements. Further
investigation is thus needed to understand how representative the chamber-based
observations from Walker et al. (2017) are of the larger-scale SPRUCE enclosures, and to
reconcile these GPP estimates with plot-level NPP observations (Norby et al., 2019).
The hydrology cycle, especially water table depth (WTD) is also a key factor that
influences the seasonality of GPP in *Sphagnum* mosses (Lafleur et al., 2005; Riutta 2007,
Sonnentag et al, 2010; Grant et al., 2012; Kuiper et al., 2014; Walker et al, 2017). One
key feedback is if the water table declines, there can be enhanced decomposition and
subsidence of the peat layer, which brings the surface down closer to the water table
again. But we currently did not consider the peat layer elevation changes in our model
and this will be one of the future development directions. The capillary rise plays into the
*Sphagnum* hydrological balance, which varies depending on water table depth and
evaporative demand. At short timescales or under rapidly changing conditions, there may
not be equilibration between the Sphagnum water content and the peat moisture.
Generally, the *sphagnum* water content will equilibrate with the peat on a daily basis
outside the plot since the dew point is often reached at night. But inside the warmer plots
since the vapor pressure deficit does not go to zero some disequilibration could remain.
High-frequency latent heat flux data from the site are currently lacking, but could help to
constrain these effects in the future. The current phenology observations also include if
*Sphagnum* hummock and hollow are wet or dry, and we could look at the relationship
with soil water content sensors in the future.  Moreover, the equilibration time between
peat moisture and moss water content is reasonable fast, but the timescales for rewetting
should change as the peat dries since the cross section for capillary rise will decline and
thus the maximum flux to the surface will decline. So, at some point, between gravity
potential and reduced hydraulic conductivity the capillarity will not satisfy evaporative
demand (Personal discussion with field expert Jeffrey Warren). Previous studies have
reported that drier and warmer future climates can lower the water table, affecting the
resilience of long-term boreal peatland carbon stocks (Limpens et al., 2008, Dise, 2009,
Frolking et al., 2011). WTD drawdown affects the net ecosystem productivity of boreal
peatlands through its effects on ecosystem respiration and GPP. The interactions between
WTD and GPP, however, vary across peatlands and influence both vascular and
nonvascular plant GPP in different ways (Lafleur et al., 2005). For instance, nonvascular
plants mostly access water in the near surface shallow peat layers. These layers, however,
can drain quickly with receding WTD and high nonvascular evaporative demand, and
thus depend on water supply through capillary rise or precipitation (Dimitrov et al., 2011,
Peichl et al., 2014, Druel et al., 2017). If recharge is not adequate, near-surface peat
desiccation occurs thereby cutting off the supply of water to Sphagnum, which
subsequently dries, leading to rapid decline in  GPP (Lafleur et al., 2005, Riutta 2008,
Sonnentag et al ., 2010, Sulman et al., 2010, Dmitrov et al., 2011, Kuiper et al., 2014,
Peichl et al., 2014). Thus, for the Sphagnum mosses desiccation occurs and the time
needed before recovery to optimum photosynthetic capacity should be taken into account
in our future work.  Alternately, under saturated conditions when the water table is close
to the *Sphagnum* surface, *Sphagnum* photosynthesizing tissue can become submerged or
surrounded by a film of water that is likely to reduce the effective LAI of the *Sphagnum*
and thus reduce photosynthesis (Walker et al., 2017). Submerged Sphagnum can take up
carbon derived from CH4 via symbiotic methanatrophs (Raghoebarsing et al., 2005), but
in any cases $CO_2$ diffusion for photosynthesis will dramatically decrease under water.
Larmola et al. (2014) also reported that the activity of oxidizing bacteria provides not
only carbon but also nitrogen to peat mosses and, thus, contributes to carbon and nitrogen
accumulation in peatlands, which store approximately one-third of the global soil carbon
pool. We currently didn't consider this kind of $CH_4$ associated carbon and nitrogen
uptake by *Sphagnum*.

The live green *Sphagnum* moss layer buffers the exchange of energy and water at

soil surface and regulates the soil temperature and moisture because of its high-water
holding capacity and the insulating effect (McFadden et al., 2003; Block et al., 2011;
Turesky et al, 2012; Park et al., 2018). Currently, we apply the same method for the
hummock and hollow *Sphagnum* water content prediction and can test the model against
the measured data when more data are available. Our model still can predict *Sphagnum*
water content differences between these microtopographies as expected, with the water
content of hollows greater than that of hummocks though. In addition, our model is able
to represent the self-cooling effect, although we do not yet have measurements available
to validate the model. The relationship of the differences between vegetation temperature
(TV) and 2m air temperature (TBOT) (TV-TBOT) and canopy evaporation for both
hummock and hollow *Sphagnum* demonstrated that the differences of TV-TBOT was
negative and the canopy evaporation had a negative relationship with TV-TBOT (Fig.
S3). Moreover, Walker et al., (2017) reported that the function of *Sphagnum* water
content to soil water content or to water table depth they used for the SPRUCE site was
empirical and may not be representative for peatland ecosystem.  To better represent the
peatland ecosystem in our model, we will eventually treat the *Sphagnum* mosses as the
"top" soil layer with a lower thermal conductivity and higher hydraulic capacity
(Beringer et al., 2001; Wu et al., 2016; Porada et al., 2016).

**5.2 Predicted warming and elevated CO$_2$ concentration response uncertainties**

Our model warming simulations suggested that increasing temperature reduced the *Picea* growth but increased the growth of *Larix* under both ambient and elevated atmospheric CO$_2$ conditions. The main reason for this model difference in response for the two tree species is that despite their similar productivity under ambient conditions, *Picea* has more respiring leaf and fine root biomass because of lower specific leaf area, longer leaf longevity, and higher fine root allocation. Therefore, warming results in a much larger increase in maintenance respiration relative to changes in NPP for *Picea* compared to *Larix* (Fig. 5 and Fig. S4). Increased tree growth and productivity in response to the recent climate warming for high-latitude forests has been reported (Myneni et al., 1997, Chen et al. 1999, Wilming et al. 2004, Chavardes, 2013). On the other hand, reductions in tree growth and negative correlations between growth and temperature also have been shown (Barber et al., 2000; Wilmking et al., 2004; Silva et al., 2010; Juday and Alix 2012; Girardin et al., 2016; Wolken et at., 2016).

Our model also predicted increasing growth of shrubs with increased temperature, similar to simulated increase in shrub cover caused mainly by warmer temperatures and longer growing seasons reported by Miller and Smith (2012) using their model LPJ-GUESS. In addition, several other modelling studies have also found increased biomass production and LAI related to shrub invasion and replacement of low shrubs by taller shrubs and trees in response to increased temperatures in tundra regions (Zhang et al., 2013; Miller and Smith, 2012; Wolf et al., 2008; Porada et al., 2016; Rydssa et al., 2017).

The responses of *Sphagnum* mosses to warming simulated by ELM_SPRUCE
showed that *Sphagnum* growth in hollows was consistently higher with increased
temperatures, where water availability was not limiting. *Sphagnum* growing on
hummocks, on the other hand, showed negative warming responses that are related to the
strong dependency on water table height. A Recent study of the same SPRUCE site
(Norby et al. 2019) had suggested that the hummock-hollow microtopography had a
larger influence on *Sphagnum* responses to warming than species-specific traits. In
addition, the previous studies had demonstrated that the most dominant mechanism of
*Sphagnum* warming response was probably through the effect of warming on depth to the
water table and water content of the acrotelm, both of them responded to increasing
temperature (Grosvernier et al., 1997; Rydin, 1985; Weltzin et al., 2001; Norby et al.,
2019).  Moreover, desiccation of capitula due to increased evaporation associated with
higher temperatures and vapor pressure deficits can reduce *Sphagnum* growth
independent of the water table depth (Gunnarsson et al., 2004).  We currently used the
same parameters for both hummock and hollow, but could consider species differences in
the future. Norby et al. (2019) investigated different *Sphagnum* species at the same site
and  reported there was no support for the hypothesis that species more adapted to dry
conditions (e.g., *S. magellanicum* and *Polytrichum* mainly on hummocks) would be more
resistant to the stress and would increase in dominance, and both hummock and hollow
*sphagnum* are declining with warming despite the differences between them.  This
declining trend may be in part due to increased shading from the shrub layer, which is
expanding with warming.

Deleted: and

Formatted: Font: Italic

Formatted: Font: Not Italic

Formatted: Font: Italic

Ecosystem warming can have direct and indirect effects on *Sphagnum* moss
growth. The growth of Sphagnum may be reduced directly by higher air temperature, due
to the relatively low temperature optima of moss photosynthesis (Hobbie et al.,1999; Van
Gaalen, 2007; Walker et al., 2017). On the other hand, increased shading by the shrub
canopy and associated leaf litter could indirectly decrease moss growth (Chapin et al.,
1995; Hobbie and Chapin 1998; Van der Wal et al., 2005; Walker et al., 2006; Breeuwer
et al., 2008). In contrast, other studies suggest that *Sphagnum* growth can be promoted
via a cooling effect of shading on the peat surface, by alleviating photo-inhibition of
photosynthesis and also by reducing evaporation stress (Busby et al., 1978; Murray et al.,
1993; Man et al., 2008; Walker et al., 2015, Bragazza et al., 2016, Mazziotta et al., 2018).
Our model sensitivity analysis also indicated that the parameters of Shrub showing
significant sensitivities to *Sphagnum* mosses GPP, indicating that competition between
the PFTs for resources might be important. Moreover, ELM_SPRUCE did predict
enhancement of shrub and *Larix* tree with increased temperatures with both ambient and
elevated $CO_2$ conditions (the leaf area increasing with warming, Fig. S3). Currently
ELM_SPRUCE does not include light competition among multiple PFTs, and thus does
not represent cross-PFT shading effects, which may contribute to the warming and
elevated $CO_2$ response differences between our model prediction and observed result of
Norby et al. (2019). Meanwhile, we have fixed cover fraction for PFTs in our model may
also contribute to the disagreement of predicted and observed warming responses. While
Norby et al. (2019) showed that the fractional cover of different *Sphagnum* species
declined with warming.
*Sphagnum* mosses are sitting on top of high $CO_2$ sources. $CH_4$ can be a significant
carbon sources of submerged *Sphagnum* (Raghoebarsing et al., 2005; Larmola et al,
2014); refixation of $CO_2$ derived from decomposition processes also is an important
source of carbon for *Sphagnum* (Rydin and Clymo, 1989; Turetsky and Wieder, 1999).
The effects of the elevation of atmospheric $CO_2$ on *Sphagnum* moss are currently
disputed, with studies indicating an increase in growth rate (Jauhiainen and Silvde 1999;
Heijmans et al. 2001a; Saarnio et al. 2003), decreases in growth rate (Grosvernier et al.
2001; Fenner et al. 2007) and no response (Van der Hejiden et al. 2000; Hoosbeek et al.
2002; Toet et al. 2006). Norby et al. (2019) indicated that no growth stimulation of both
hummock and hollow *Sphagnum* under elevated $CO_2$ condition, but significant negative
effects of elevated $CO_2$ on *Sphagnum* NPP in year 2018 at the same study site.
Contrasting responses between *Sphagnum* species are thought to be coupled with the
water availability. In contrast, our model results showed that both hummock and hollow
*Sphagnum* growths were stimulated by the elevated $CO_2$ concentration, which may be
attributed to the fact thatvwe did not consider the light competition between the PFTS
(shrub and tree shading effects) and use a fixed cover fraction of *Sphagnum*.
The $CO_2$ vertical concentration profile is assumed to be uniform in the
simulations. In the experiment, the enclosure's regulated additions of pure $CO_2$ are
distributed to a manifold that splits the gas into four equal streams feeding each of the
four air handling units (Hanson et al., 2017 Fig. 2a), and is injected into the ductwork of
each furnace just ahead of each blower and heat exchanger. Horizontal and vertical
mixing within each enclosure homogenizes the air volume distributing the $CO_2$ along
with the heated air. The horizontal blowers in the enclosures together with external wind
eddies ensure vertical mixing.  We do not have routine automated $CO_2$ concentration data
below 0.5m.  The moss layer may well be experiencing higher concentrations than
assumed by the model, but such an impact will be minimized during daylight hours.
Preliminary isotopic measurements imply a significant fraction of carbon assimilated by
the moss may come from subsurface respired $CO_2$ (i.e., $CO_2$ with older $^{14}C$ signatures
predating bomb carbon that can only be sourced from deeper peat, Hanson et al., 2017).
However, the observed elevated $CO_2$ response is smaller than simulated (Hanson et al.,
2020). Understanding the drivers of elevated $CO_2$ response or lack thereof is a key topic
for future work and we will consider this effect in future assessments of the isotopic
carbon budgets for the SPRUCE study.
To better investigate the *Sphagnum* warming and elevated $CO_2$ responses, we should
also focus on revealing the interactions with Shrub and nitrogen availability (Norby et al.,
2019). Nitrogen ($N_2$) fixation is a major source of available N in ecosystems that receive
low amounts of atmospheric N deposition, like boreal forests and subarctic tundra (Lindo
et al., 2013, Weston et al, 2015, Rousk et al., 2016, Kostka et al., 2016). For example,
diazotrophs are estimated to supply 40-60% of N input to peatlands (Vile et at., 2014)
with high accumulation of fixed N into plant biomass (Berg et al., 2013). Nevertheless,
$N_2$ fixation is an energy costly process and is inhibited when N availability and reactive
nitrogen deposition is high (Gundale et al., 2011; Ackermann et al., 2012; Rousk et al.,
2013). This could limit ecosystem N input via the $N_2$ fixation pathway. We are measuring
Sphagnum associated N2 fixation at the SPRUCE site and found that rates decline with
increasing temperature (Carrell et al. 2019 Global Change Biology). We are continuing
these measurements to see if they correlate with the GPP empirical relationship from

**Formatted:** Subscript

**Formatted:** Superscript

**Formatted:** Subscript

**Formatted:** Subscript

**Formatted:** Font color: Red

**Moved up [1]:** ELM_SPRUCE does predict enhancement of shrub and Larix tree with increased temperatures with both ambient and elevated $CO_2$ conditions (the leaf area increasing with warming, Fig. S3).  Norby et al. (2019) showed that the fractional cover of different Sphagnum species declined with warming, but while ELM_SPRUCE allows the canopy density of PFTs to change prognostically,

**Deleted:** ELM_SPRUCE does predict enhancement of of shrub and Larix tree with increased temperatures with both ambient and elevated $CO_2$ conditions (the leaf area increasing with warming, Fig. S3).  Norby et al. (2019) showed that the fractional cover of different Sphagnum species declined with warming, but while ELM_SPRUCE allows the canopy density of PFTs to change prognostically, their fractional cover is held constant.

**Formatted:** Font: Italic

**Formatted:** Subscript

Clevand (1999), or if temperature disrupts that association. Once finished, results will be
used to represent N fixation by the Sphagnum layer and testing with measurements.
It is also encouraging that while we did not use leaf-level gas exchange
observations in our optimization, the increased maintenance respiration base rate and
temperature sensitivity compared to default (table 2) is largely consistent with pre-
treatment leaf level observations (Jensen et al., 2019). In the future, a multi-scale
optimization framework that can assimilate leaf and plot-level observations
simultaneously should lead to improved model predictions and reduced uncertainties for
the treatment simulations. If similar patterns observed in ambient conditions continue
during the treatments, incorporating seasonal variations in leaf photosynthetic parameters
may also further improve the simulated response to warming (Jensen et al., 2019).
Overall, while the sensitivity analysis is useful to indicate the key parameters and
mechanisms responsible for uncertainty, our ability to quantify prediction uncertainty is
limited because we consider only a single simulation with optimized parameters. Ideally,
we should perform a model ensemble that represents the full range of posterior
uncertainty over simulations that are consistent with the pre-treatment observations, and
also a range of possible future meteorological conditions. This is currently being done for
SPRUCE with the TECO carbon cycle model (Jiang et al., 2018), but the computational
expense of ELM_SPRUCE currently prohibits this approach. By combining new
surrogate modeling approaches (e.g. Lu et al., 2019) with MCMC techniques, it may be
possible to achieve this in the near future. This will help to reduce prediction
uncertainties, which currently prevail in the future carbon budget of peatlands and its
feedback to climate change (McGuire et al., 2009).
The algorithms used to represent moss (e.g. Williams and Flanagan) are
transferable to and have been applied by other modeling groups in other
peatlands.  However, we expect that certain parameters will vary, for example, the
microtopographic parameters, the relationship between peat moisture and internal water
content, and moss properties such as C:N ratio.  The parameter sensitivity analysis
informs us as to the most important parameters responsible for prediction uncertainty,
and can inform how to prioritize these measurements.  Collecting these measurements
from a variety of sites will be a necessary preliminary exercise. In addition to the
simulations aimed at improved understanding of bog response to experimental
manipulations at the plot-scale, we are pursuing model implementations at larger spatial
scales. The model framework described in this study is capable of performing regional
simulations, although the current simulations were designed for mechanistic
understanding of *Sphagnum* mosses hydrological and physiological dynamics at the plot-
level.

**6. Summary**
In this study, we reported the development of a *Sphagnum* moss PFT and
associated processes within the ELM_SPRUCE model. Before being used to examine the
ecosystem response to warming and elevated $CO_2$ at a temperate bog ecosystem, the
updated model was evaluated against the observed *Sphagnum* GPP and annual NPP,
aboveground tree biomass and shrub stem biomass. The new model can capture the
seasonal dynamics of moss *Sphagnum* GPP, but with lower peak GPP compared to site-
level observations, and can predict reasonable annual values for *Sphagnum* NPP but with
lower interannual variation. Our model largely agrees with observed tree and shrub
biomass. The model predicts that different PFTs responded differently to warming levels
under both ambient and elevated $CO_2$ concentration conditions. The NPP of the two
dominant tree PFTs (black spruce and *Larix*) showed contrasting responses to warming
scenarios (increasing with warming for *Larix* but decreasing for black spruce), while
shrub NPP had similar warming response to *Larix*. Hummock and hollow *Sphagnum*
showed opposite warming responses: hollow *Sphagnum* shows generally higher growth
with warming, but the hummock *Sphagnum* demonstrates more variability and strong
dependence with water table height. The ELM predictions further suggest that the effects
of $CO_2$ fertilization can change the direction of the warming response for the bog
peatland ecosystem, though observations of *Sphagnum* species at the site does not yet
appear to support this (Norby et al. 2019).
Data availability. The model code we used is available here:
https://github.com/dmricciuto/CLM_SPRUCE. The datasets and scripts were used for the figures
is here: https://github.com/dmricciuto/CLM_SPRUCE/tree/master/analysis/Shietal2020

**Acknowledgements**
Research was supported by the U. S. Department of Energy, Office of Science,
Biological and Environmental Research Program. Oak Ridge National Laboratory is
managed by UT-Battelle, LLC, for the US Department of Energy under contract DE-
AC05–00OR22725.

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
