# Peer review of "Extending a land-surface model with Sphagnum moss to simulate responses of a northern temperate bog to whole-ecosystem warming and elevated $CO_2$"

_Biogeosciences, 2020_

## Referee Comment (RC1) · Anonymous Referee #1 · 27 May 2020

Even though, mosses are ubiquitous part of boreal vegetation, especially so in peatlands, mosses and their contribution to ecosystem functions are overlooked. The study introduces new plant functional type (PFT) with Sphagnum-specific processes that can be in some extent to be used to describe mosses in other boreal and arctic environments, e.g. upland forests and wet tundra. Manuscript consists, sensitivity analysis of updated land model component and validation part that takes place in boreal ombrotrophic, raised-dome bog peatland with warming and $CO_2$ enrichment experiment.

Authors have stated that drier and warmer future climates can lower water table and it has implications on growth of Sphagnum. In the study, capillary rise is a function of peat water content in 10 cm, but it is not clearly stated that if measured or modelled values of peat water content is used in sensitivity analysis and in case study and how

water table fluctuations affect functions (e.g. gross primary production) of Sphagnum mosses? If there is more Larix and taller shrubs growing on the site, does it drain more or less the site? Do you see effect of warmer climate on water table depth in higher temperatures and how this will affect Sphaghnum mosses?

What I am after is that which kind of hydrological feedbacks there are and how it affects ecophysiology of Sphagnum PFT if temperature will increase +9.0 degrees of Celsius. This is something to think about especially if capillary connection of Sphagnum is described through a simple relationship between capitulum water content and peat water content at 10 cm depth. This could be answered simply by studying hydrological balance of Sphagnum PFT and showing how large part capillary rise plays in Sphagnum hydrological balance. Is it even necessary or which kinds of implications it has to photosynthetic capacity or other ecophysiological processes? In my opinion, authors have not clearly showed or discussed underlying assumptions and consequences of made choices and it should be improved.

Sphagnum mosses are sitting on top of high CO2 (and water vapor) sources and experiencing naturally higher concentrations of CO2. How this affects to gross primary production of mosses and which kind of differences there possibly are between mosses that are located to hollows and hummocks? How does this fit to CO2 enrichment study? Is CO2 concentration profile assumed to be uniform throughout the canopy profile? Does this have effects on results of simulations?

In Chapter 5.3 authors raise important issues and future directions. To me problem is that now it seems to be detached from the model description and discussions. Could this be embedded better in discussions to make the manuscript more coherent and structure clearer?

L183: Are measurements of Sphagnum water contents from Sphagnum growing on hollows and hummocks? Were there any differences between these microtopograpichal features on water content in moss? Even though, this is clever way to solve

capillary rise issue of mosses in simple manner but is this method applicable in both microtopographical positions? My main concern is that does this approach mask the effects of hydrology that is quite important for Sphagnum ecophysiology (main source of water in hollows and hummocks). How about self-cooling (enhanced evaporation) of Sphagnum covered surfaces due to capillary rise? Does the static approach fail especially in sunny days and which kinds of implications it has to Sphagnum ecophysiology?

L589-L591 This is not only in case with submerged Sphagnum, but it seems that Sphagnum utilizes $CH_4$ as an indirect source of $CO_2$ (e.g. Larmola et al., 2014: DOI: 10.1073/pnas.1314284111)

L614-618: Can it be that model parameters of hollow and hummock Sphagnum can differ from each other? How this could affect outcome of simulations? I would quess that Sphagnum growing on hummocks are more drought tolerant and resistant than those species growing in hollows. This could be seen i.e. in different slatop -value and, as discussed by authors, in base rate for maintenance respiration.

L684: Is N fixation somehow represented in a model. Should that be mentioned in a model description? In my opinion, this is quite interesting and important part why moss PFTs should be included in models handling boreal and arctic regions.

---

## Referee Comment (RC2) · Samuli Launiainen (Referee) · 18 Jun 2020

**Review of bg-2020-90 Shi et al: Modeling the hydrology and physiology of Sphagnum moss in a northern temperate bog**

In this study, new plant functional type (PFT) describing Sphagnum –moss, abundant in boreal and arctic peatlands, is incorporated into the land model component ELM of the Earth System model E3SM to better represent carbon, water and nutrient cycling in boreal and arctic regions. The ELM with the newly proposed Sphagnum-PFT was parameterized and evaluated against data collected under ambient conditions and under climate-change experiment conducted at an ombrothrophic bog in Minnesota, US. Further, the model is used to predict changes in moss and vascular plant productivity, biomass accumulation and water table level for combined temperature and $CO_2$ increase scenarios.

The article is well written and fits topically under the scope of BG. However, the relevance for larger scientific community is limited as the study is centered on development of a specific land model and testing for a specific site. I therefore consider the study as borderline case for BG and maybe fits better into a more specific model development journal such as Geoscientific Model Development. However, this is up to the Editor to decide.

In recent years there has been strong interest on including Sphagnum as well as feather mosses and other bryophytes into land-surface models. In addition to the references listed in the Introduction the authors should take a look and cite the recent works of Philip Porada and colleagues (Porada et al., 2013, 2016), as well as note the inclusion of moss-PFT into ORCHIDEE-model (Druel et al., 2017). The authors should also be more explicit how their study builds on and improves the existing knowledge and methods to describe Sphagnum mosses in land surface models. If the study is to be published in BG, the results and methods should in my opinion be generalized and better interpreted against existing literature. Currently, the discussion, in particular Section 5.3, reads more as a research plan for future development of a specific land surface model.

**My general comments are as follows:**

1)  Modeling Sphagnum water content

Sphagnum total water content is sum of two pools: $W_{tot} = W_{internal} + W_{surface}$

There are few remarks / comments that should be made.

First, $W_{internal}$ is described as non-linear function of top soil water content and thus immediately adjust to changes in soil water content (or water table). This approach thus assumes that in Sphagnum, $W_{internal}$ is at hydrostatic equilibrium with soil water potential (or water content) as defined through water-retention characteristics of the peat-Sphagnum continuum. Moreover, it assumes that hydraulic conductivity is sufficiently large so that Sphagnum water content is never decoupled from soil water content. Such assumptions may not hold in case water table (WT) drops deep during prolonged dry periods, more propable in future climate.

What is author's conclusion on generality of $W_{internal}$ – soil water content relationship (Fig. 1) among Sphagnum species (hummock vs. hollow –preferences)? And how $W_{internal}$ and $W_{external}$

pools were separated from the gravimetric measurements of water content in Sphagnum to derive relationship between Winternal and soil water content?

Second, the Wsurface is filled by interception of rainfall (how about condensation?) and drained by evaporation. I wonder how the surface storage capacity is described and parameterized and whether Wsurface and Winteral are completely independent water pools? See also Porada et al. (2018).

Third, the authors should describe how evapo(transpi)ration from Sphagnum-PFT is modeled and how it differs from vascular-PFT's. From which water pools evaporation takes place and how evaporation rate or surface conductance depend on Sphagnum characteristics and near-ground microclimate. How and whether evaporation is restricted with decreasing water content? This is required to understand e.g. how SLA and leaf C:N ratio can affect evapotranspiration and interpret the results of sensitivity analysis in Fig 2.

2) Modeling Sphagnum photosynthesis

Standard Farquhar-approach is used to simulate Sphagnum net $CO_2$ demand given air-chloroplast conductance described as non-linear function of Wtot (eq. 6, from Williams and Flanagan, 1998). In addition, submerging of Sphagnum is assumed to 'kill' $CO_2$ diffusion and thus a restriction to photosynthetic uptake is applied and described as linear function of submerged to total photosynthesizing height (here 0.05m) of the moss. Does the implementation of moss photosynthesis follow Walker et al. (2017)?

I like the approach but wonder whether the relatively poor match between modeled and 'measured' moss GPP (Fig 3) can be due i) to ill-represented or omitted temperature response or seasonal acclimation of Vcmax etc., ii) biased Sphagnum temperature (how was it modeled – from surface energy balance?) or ii) too strong submerge-impact. As Sphagnum moss has high leaf (or shoot) area, radiation decays rapidly with canopy depth and thus the top centimeter(s) of the shoot system are responsible of majority of photosynthetic activity. For instance, Niinemets and Tobias (2014) and Zotz and Kahler (2007) show light attenuation profiles and photosynthesis profiles for some moss species. Considering typical characteristics (color) of Sphagnum-canopy, assuming $CO_2$ uptake is evenly distributed across top 5cm may lead to overestimated submerge-impact.

I also wonder whether the soil-respired $CO_2$ leads the Sphagnum to operate in $CO_2$ enriched atmosphere already in current conditions and whether this would lead to over-estimated increase of GPP at 900 ppm as photosynthetic $CO_2$-response curve has saturating shape?

In results L530-534 it is stated that modeled Sphagnum biomass correlates with water table and best correlation is found at with 3-month timelag. For GPP and NPP the instantaneous dependence on WT is from Fig 1. and eq. 6. Please describe how NPP is allocated into biomass and how the growth dynamics of Sphagnum-PFT is modeled; can this explain the timelag?

3) Modeled carbon cycle components and responses to warming and elevated CO2

For the reader to understand the modeled carbon cycle responses, it is necessary that ELM 'tiling-scheme', pathway from NPP to biomass growth and between-PFT competition are better described

in Section 2.1 and/or 3.3. That is, present information such as L627-634 earlier in the manuscript. Are shrubs and Sphagnum present as independent tiles or do they occur below the overstory trees?

4) Title: "Modeling the hydrology and physiology of Sphagnum moss in a northern temperate bog" should be revised to match the manuscript content. The study is on extending the land-surface model with Sphagnum-PFT and simulating response of moss and vascular vegetation productivity to warming and increasing atmospheric CO2.

**Specific comments:**

L98: water and exchanges within peatland and between peatland and atmosphere?

L 146-147: new chapter – study Aims.

L178-179: Evaporation depends on evaporative demand (VPD; available energy), moss-atmosphere conductance (moss canopy structure, roughness and flow characteristics) and available water pool. The latter is then depends on capillary rise from water table.

L196: canopy_water → can_water

L211: eq. 6 uses total water content, not Winternal

L238-239: this assumes boundary-layer conductance >> moss surface – chloroplast conductance; assumption is ok but could be mentioned. Note also that maximum g_tc may vary among Sphagnum species?

L284: what is pre-treatment data?

L363-367: please elaborate whether the data used in parameter optimization is independent of data used in model testing (Fig. 3-4)

L393: point should be (*)

L479-480: Just curious - why year 2012 was an exception? Were env. drivers different?

L522: Fig. 5: what is driving the strong inter-annual variability of Sphagnum and shrub NPP (annual variability has different sign among these PFT's). Is this mainly due to WT height and does root zone water content affect vascular PFT photosynthesis (O2-stress in wet conditions)?

L616-618: this is quite trivial result as Sphagnum water content was made proportional to soil water content (and hence WT).

L659: The question is that to which extent the parameterization from S1-Bog be generalized to other peatlands?

L667: See e.g. Beringer et al. (2001) and Porada et al. (2016) who have already done this.

**References:**

Beringer, J., Lynch, A.H., Chapin III, F.S., Mack, M. and Bonan, G.B., 2001. The representation of arctic soils in the land surface model: the importance of mosses. Journal of Climate, 14(15), pp.3324-3335.

Druel, A., Peylin, P., Krinner, G., Ciais, P., Viovy, N., Peregon, A., Bastrikov, V., Kosykh, N. and Mironycheva-Tokareva, N., 2017. Towards a more detailed representation of high-latitude vegetation in the global land surface model ORCHIDEE (ORC-HL-VEGv1. 0). Geoscientific Model Development, 10(12), p.4693.

Niinemets, Ü. and Tobias, M., 2014. Scaling light harvesting from moss "leaves" to canopies. In Photosynthesis in Bryophytes and Early Land Plants (pp. 151-171). Springer, Dordrecht.

Porada P, Van Stan II J, Kleidon A (2018) Significant contribution of non-vascular vegetation to global rainfall interception. Nature Geoscience, doi: 10.1038/s41561-018-0176-7

Porada P, Ekici A, Beer C (2016) Effects of bryophyte and lichen cover on permafrost soil temperature at large scale The Cryosphere 10, 2291-2315

Porada P, Weber B, Elbert W, Pöschl U, Kleidon A (2013) Estimating global carbon uptake by Lichens and Bryophytes with a process-based model. Biogeosciences 10, 6989-7033

Walker, A. P., Carter, K. R., Gu, L., Hanson, P. J., Malhotra, A., Norby, R.J., Sebestyen, S. D., Wullschleger, S. D., Weston, D. J.: 2017. Biophysical drivers of seasonal variability in Sphagnum gross primary production in a northern temperate bog, J. Geophys. Res.-Biogeo, 122, 1078-1097, https://doi.org/10.1002/2016JG003711, 2017.

Zotz, G. and Kahler, H., 2007. A moss "canopy"–Small-scale differences in microclimate and physiological traits in Tortula ruralis. Flora-Morphology, Distribution, Functional Ecology of Plants, 202(8), pp.661-666.

---

## Author Comment (AC1) · 29 Aug 2020

Reviewer 1

Even though, mosses are ubiquitous part of boreal vegetation, especially so in peatlands, mosses and their contribution to ecosystem functions are overlooked. The study introduces new plant functional type (PFT) with Sphagnum-specific processes that can be in some extent to be used to describe mosses in other boreal and arctic environments, e.g. upland forests and wet tundra. Manuscript consists, sensitivity analysis of updated land model component and validation part that takes place in boreal ombrotrophic, raised-dome bog peatland with warming and $CO_2$ enrichment experiment. Authors have stated that drier and warmer future climates can lower water table and

it has implications on growth of Sphagnum. In the study, capillary rise is a function of peat water content in 10 cm, but it is not clearly stated that if measured or modelled values of peat water content is used in sensitivity analysis and in case study and how water table fluctuations affect functions (e.g. gross primary production) of Sphagnum mosses?

We use modeled values of peat water content in both the forward simulations and the sensitivity analysis. The modeled water table (WT)WT has a direct effect on the sphagnum GPP in our model when WT is above the soil surface, through submergence effects (manuscript Eq 7). WT also has an indirect effect on sphagnum GPP, through total conductance to CO2 (gtc), as follows: gtc increases with total sphagnum water content (manuscript Eq. 6) while GPP increases with gtc (manuscript Eqs. 5 and 4). Total sphagnum water content includes a component from sphagnum internal water (manuscript Eq 3) which is an empirically derived function of soil water nearest the 10cm soil horizon (manuscript Eq 2). As WT drops below the 10cm soil horizon the water content in that layer declines, leading to lower sphagnum internal water, lower sphagnum total water, lower gtc, and lower GPP.

If there is more Larix and taller shrubs growing on the site, does it drain more or less the site? Do you see effect of warmer climate on water table depth in higher temperatures and how this will affect Sphaghnum mosses?

From manuscript Fig.6, we can see the relative biomass changes of Larix, shrub and hollow Sphagnum increase with temperature, the hummock Sphagnum biomass decreases with warming and is more dependent with water table height, and the water table generally decreases with temperature. We have added "We plotted the predicted canopy evaporation for hummock and hollow Sphagnum responses to warming and found that both hummock and hollow Sphagnum canopy evaporation increase with warming for both ambient and elevated atmospheric CO2 conditions despite the Larix and shrubs are growing with warming. Moreover, the hollow Sphagnum canopy evaporation warming response is stronger than that of the hummock Sphagnum (Fig. S2)."

to the text (L615-620).

What I am after is that which kind of hydrological feedbacks there are and how it affects ecophysiology of Sphagnum PFT if temperature will increase +9.0 degrees of Celsius. This is something to think about especially if capillary connection of Sphagnum is described through a simple relationship between capitulum water content and peat water content at 10 cm depth. This could be answered simply by studying hydrological balance of Sphagnum PFT and showing how large part capillary rise plays in Sphagnum hydrological balance. Is it even necessary or which kinds of implications it has to photosynthetic capacity or other ecophysiological processes? In my opinion, authors have not clearly showed or discussed underlying assumptions and consequences of made choices and it should be improved.

To make how the hydrological cycle affects the Sphagnum ecophysiology clear, we have added "One key feedback is if the water table declines, there can be enhanced decomposition and subsidence of the peat layer, which brings the surface down closer to the water table again. But we currently did not consider the peat layer changes in our model and this will be one of the future development directions. The capillary rise plays in Sphagnum hydrological balance, which varies depending on water table depth and evaporative demand. The sphagnum water content will equilibrate with the peat on a daily basis outside the plot since the dew point is reached. But inside the plots since the VPD does not go to zero there could remain some disequilibration. The current phenology observations include if sphagnum hummock and hollow are wet or dry, and we could look at the relationship with soil water content sensors at some point." to the text (L 668-679). We also added one more reference " Druel et al., 2017" to L693", "Thus, for the Sphagnum mosses desiccation occurs and the time needed before recovery to optimum photosynthetic capacity should be taken into account in our future work" to L 697-699, and "Larmola et al. (2014) also reported that the activity of oxidizing bacteria provides not only carbon but also nitrogen to peat mosses and, thus, contributes to carbon and nitrogen accumulation in peatlands, which store approximately one-third of

the global soil carbon pool. We currently didn't consider this kind of CH4 associated carbon and nitrogen uptake by Sphagnum" to L705-709. We will eventually treat the Sphagnum mosses as the "top" soil layer with a lower thermal conductivity and higher hydraulic capacity than a mineral soil layer.

Sphagnum mosses are sitting on top of high CO2 (and water vapor) sources and experiencing naturally higher concentrations of CO2. How this affects to gross primary production of mosses and which kind of differences there possibly are between mosses that are located to hollows and hummocks? How does this fit to CO2 enrichment study?

To clarify the elevated CO2 concentration responses of Sphagnum, we add "Sphagnum mosses are sitting on top of high CO2 sources. CH4 can be a significant carbon sources of submerged Sphagnum (Raghoebarsing et al., 2005; Larmola et al, 2014); refixation of CO2 derived from decomposition processes also is an important source of carbon for Sphagnum (Rydin and Clymo, 1989; Turetsky and Wieder, 1999). The effects of the elevation of atmospheric CO2 on Sphagnum moss are currently disputed, with studies indicating an increase in growth rate (Jauhiainen and Silvde 1999; Heijmans et al. 2001a; Saarnio et al. 2003), decreases in growth rate (Grosvernier et al. 2001; Fenner et al. 2007) and no response (Van der Hejiden et al. 2000; Hoosbeek et al. 2002; Toet et al. 2006). Norby et al. (2019) indicated that no growth stimulation of both hummock and hollow Sphagnum under elevated CO2 condition at the same study site. There are, however, significant negative effects of elevated CO2 on Sphagnum NPP in year 2018. Contrasting responses between Sphagnum species are thought to be coupled with the water availability. In contrast, our model results showed that both hummock and hollow sphagnum growths were stimulated by the elevated CO2 concentration, which may attribute to we did not consider the light competition between the PFTS (shrub and tree shading effects) and use the fixed cover fraction of Sphagnum." to the main text L831-846.

Is CO2 concentration profile assumed to be uniform throughout the canopy profile? Does this have effects on results of simulations?

We added "The CO2 concentration profile is assumed to be uniform in the simulations. In the experiment, the enclosure's regulated additions of pure CO2 are distributed to a manifold that splits the gas into four equal streams feeding each of the four air handling units (Hanson et al., 2017 Fig. 2a), and is injected into the ductwork of each furnace just ahead of each blower and heat exchanger. Horizontal and vertical mixing within each enclosure homogenizes the air volume distributing the CO2 along with the heated air. The horizontal blowers in the enclosures together with external wind eddies ensure vertical mixing. We do not have routine automated CO2 concentration data below 0.5m. The moss layer may well be experiencing higher concentrations than assumed by the model, but such an impact will be minimized during daylight hours. Preliminary isotopic measurements imply a significant fraction of carbon assimilated by the moss may come from subsurface respired CO2 (i.e., CO2 with older 14C signatures predating bomb carbon that can only be sourced from deeper peat, Hanson et al. 2017). We will consider this effect in future assessments of the isotopic C budgets for the SPRUCE study." to L 847-863.

In Chapter 5.3 authors raise important issues and future directions. To me problem is that now it seems to be detached from the model description and discussions. Could this be embedded better in discussions to make the manuscript more coherent and structure clearer?

Thank you for your good suggestion. We embedded 5.3 Section to Section 5.1 and 5.2 and changed Section 5.2 from "Predicted warming response uncertainties" to "Predicted warming and elevated CO2 concentration response uncertainties".

L183: Are measurements of Sphagnum water contents from Sphagnum growing on hollows and hummocks? Were there any differences between these microtopograpichal features on water content in moss? Even though, this is clever way to solve capillary rise issue of mosses in simple manner but is this method applicable in both microtopographical positions? My main concern is that does this approach mask the effects of hydrology that is quite important for Sphagnum ecophysiology (main source

of water in hollows and hummocks). How about self-cooling (enhanced evaporation) of Sphagnum covered surfaces due to capillary rise? Does the static approach fail especially in sunny days and which kinds of implications it has to Sphagnum ecophysiology?

The measurements of Sphagnum water content during sensor calibrations were primarily on hummock species but included some hollow species. They were not separated during measurements since we needed an integrated measurement for reference against the automated subsurface sensors. We have added this information to the water content dynamics of Sphagnum mosses Section (L215-220). "Currently, we apply the same method for both hummock and hollow Sphagnum water content prediction, and can test the model against the measured data when more data are available. We do see the model predicted Sphagnum water content differences between these two microtopographies as expected, with the water content of hollows greater than that of hummocks. ELM is able to represent the self-cooling effect, although we do not yet have measurements available to validate the model. We looked at vegetation temperature (TV) - 2m air temperature (TBOT) as a function of canopy evaporation for both hummock and hollow Sphagnum, the differences of TV-TBOT is negative and the canopy evaporation has a negative relationship with TV-TBOT (Fig. S3)." has been added to L718-739.

L589-L591 This is not only in case with submerged Sphagnum, but it seems that Sphagnum utilizes CH4 as an indirect source of CO2 (e.g. Larmola et al., 2014: DOI: 10.1073/pnas.1314284111)

We added "Larmola et al. (2014) also reported that the activity of methane oxidizing bacteria provides not only carbon but also nitrogen to peat mosses and, thus, contributes to carbon and nitrogen accumulation in peatlands, which store approximately one-third of the global soil carbon pool." to the manuscript text and cited this literature (L705-709). We also cited it as reference in L832-833.

L614-618: Can it be that model parameters of hollow and hummock Sphagnum can

differ from each other? How this could affect outcome of simulations? I would quess that Sphagnum growing on hummocks are more drought tolerant and resistant than those species growing in hollows. This could be seen i.e. in different slatop -value and, as discussed by authors, in base rate for maintenance respiration.

To clarify these aspects we added "We currently use the same parameters for both hummock and hollow, but could consider species differences in the future. Norby et al. (2019) investigated the same site Sphagnum species and reported the decline in Sphagnum cover affected both S. angustifolium/fallax and S. magellanicum, but the relative loss of S. magellanicum was greater, S. magellanicum, which was present primarily on hummocks and has morphological adaptations generally favorable for drier conditions, comprised a smaller fraction of the Sphagnum community in enclosures exposed to the warmer temperatures. However, the dominant response was a sharp decline in abundance of both species, and there was no evidence that S. angustifolium/fallax was replacing S. magellanicum. There was no support for the hypothesis that species more adapted to dry conditions (e.g., S. magellanicum and Polytrichum) would be more resistant to the stress and would increase in dominance. Despite these differences, both hummock and hollow sphagnum are declining with warming. This declining trend may be in part due to increased shading from the shrub layer, which is expanding with warming." to text L 787-801. ELM is currently not able to represent this shading effect and we will address this in future model development.

L684: Is N fixation somehow represented in a model. Should that be mentioned in a model description? In my opinion, this is quite interesting and important part why moss PFTs should be included in models handling boreal and arctic regions.

We added "Inputs of new mineral nitrogen of ELM are from atmospheric deposition and biological nitrogen fixation. The fixation of new reactive nitrogen from atmospheric N2 by soil microorganisms is an important component of nitrogen budgets. ELM simply follows Cleveland et al. (1999) suggested empirical relationship that biological nitrogen fixation is a function of net primary production to predict the nitrogen fixation" to Section

2.1(L173-178). We also added "We are measuring Sphagnum associated N2 fixation at the SPRUCE site and found that rates decline with increasing temperature (Carrell et al. 2019 Global Change Biology). We are continuing these measurements to see if they correlate with the GPP empirical relationship from Clevand (1999), or if temperature disrupts that association. Once finished, results will be used to represent N fixation by the Sphagnum layer and testing with measurements." to L889-894
* * *

---

## Author Comment (AC2) · 29 Aug 2020

Reviewer 2

In this study, new plant functional type (PFT) describing Sphagnum –moss, abundant in boreal and arctic peatlands, is incorporated into the land model component ELM of the Earth System model E3SM to better represent carbon, water and nutrient cycling in boreal and arctic regions. The ELM with the newly proposed Sphagnum-PFT was parameterized and evaluated against data collected under ambient conditions and under climate-change experiment conducted at an ombrothrophic bog in Minnesota, US. Further, the model is used to predict changes in moss and vascular plant productivity, biomass accumulation and water table level for combined temperature and CO2

increase scenarios. The article is well written and fits topically under the scope of BG. However, the relevance for larger scientific community is limited as the study is centered on development of a specific land model and testing for a specific site. I therefore consider the study as border line case for BG and maybe fits better into a more specific model development journal such as Geoscientific Model Development. However, this is up to the Editor to decide.

The primary goal of the SPRUCE project is to test how vulnerable an important Carbon-rich terrestrial ecosystem is to atmospheric and climatic change by warming the entire soil profile and measuring whether large amounts of $CO_2$ and $CH_4$ are emitted The regression design allows the derivation of key temperature response functions for mechanistic ecosystem processes that can be used for model validation and improvement. In this study, we introduce a moss PFT into the land model component (ELM) of the Energy Exascale Earth System Model (E3SM). Then, we evaluate our updated model against numerous measurements. We also apply the updated ELM to explore how an ombrotrophic, raised-dome bog peatland ecosystem will respond to different scenarios of warming and elevated atmospheric $CO_2$ concentration. The model development is only part of our goal, and we mainly focus on using the model to investigate the peatland ecosystem responses to changing climate and the feedbacks.

In recent years there has been strong interest on including Sphagnum as well as feather mosses and other bryophytes into land-surface models. In addition to the references listed in the Introduction the authors should take a look and cite the recent works of Philip Porada and colleagues (Porada et al., 2013, 2016), as well as note the inclusion of moss-PFT into ORCHIDEE-model (Druel et al., 2017). The authors should also be more explicit how their study builds on and improves the existing knowledge and methods to describe Sphagnum mosses in land surface models. If the study is to be published in BG, the results and methods should in my opinion be generalized and better interpreted against existing literature. Currently, the discussion, in particular Section 5.3, reads more as a research plan for future development of a specific land

surface model.

We had added "Druel et al. (2017) investigated the vegetation-climate feedbacks in high latitudes by implementing the nonvascular plants including bryophytes and lichens to the global land surface model ORCHIFEE. Porada et al. (2016) integrated a stand-alone dynamic non-vascular vegetation model LiBry (Porada et al., 2013) to land surface scheme JSBACH, but LiBry and JSBACH mainly represent bryophyte and lichen growth on upland forest floor sites, not for wetland sites. Chadburn et al. (2015) introduced a new moss PFT to JULES land surface model and treated the thermal conductivity of moss depending on its water content." to the introduction Section and cited these literatures (L137-144). We also added a new section '2.2 Non-vascular plants: Sphagnum mosses' to Section 2 Model description to describe more details how we implement our Sphagnum mosses into our model (L184-199). For the future model development Section 5.3, we embedded into Section 5.1 and 5.2 as the other reviewer also suggested.

My general comments are as follows: 1) Modeling Sphagnum water content Sphagnum total water content is sum of two pools: Wtot = Winternal + Wsurface There are few remarks / comments that should be made. First, Winternal is described as non-linear function of top soil water content and thus immediately adjust to changes in soil water content (or water table). This approach thus assumes that in Sphagnum, Winternal is at hydrostatic equilibrium with soil water potential (or water content) as defined through water-retention characteristics of the peat-Sphagnum continuum. Moreover, it assumes that hydraulic conductivity is sufficiently large so that Sphagnum water content is never decoupled from soil water content. Such assumptions may not hold in case water table (WT) drops deep during prolonged dry periods, more propable in future climat

The equilibration time between peat moisture and moss water content is reasonable fast, but the timescales for rewetting should change as the peat dries since the cross section for capillary rise will decline and thus the maximum flux to the surface will

decline. So at some point, between gravity potential and reduced hydraulic conductivity the capillarity will not satisfy evaporative demand. But for the simplicity, we currently used the empirical representation of water content in our model for both hummock and hollow Sphagnum (as we responded above for the first viewer's comment). The function of Sphagnum water content to soil water content or to water table depth used by Walker et al. (2017) for the same site was empirical and may not be representative for the peatland ecosystem. We will treat Sphagnum as the top soil layer to allow water movement to occur along pressure gradients and thus consistently simulate Sphagnum water content. These related contents have been added to the discussion Section 5.1 L 678-739.

What is author's conclusion on generality of Winternal – soil water content relationship (Fig. 1) among Sphagnum species (hummock vs. hollow –preferences)? And how Winternal and Wexternal pools were separated from the gravimetric measurements of water content in Sphagnum to derive relationship between Winternal and soil water content?

During the calibrations, we used intact monoliths collected from multiple locations. The monoliths included both hummock and hollow species, but they were not separated during destructive measurements, since we needed an integrated measurement for comparison against the subsurface soil water content sensors. We have clarified this information to the water content dynamics of Sphagnum mosses Section (L215-220). "There are large differences in the density and traits of the different species and microtopography that would result in different relationships with soil water content. This is a difficult problem since the heterogeneity of the hummock hollow ecosystem is so great. Even so, our calibrations and measurements represent a strong effort to help reduce the uncertainty. Other efforts have not been as successful, including using remote sensed water band index and destructive surface sampling for stable water isotopes. We continue to explore new non-destructive measurements, including leaf wetness sensors, and hope to refine the measurements as the project evolves." has

been added to discussion Section 5.1 L723-731.

Second, the Wsurface is filled by interception of rainfall (how about condensation?) and drained by evaporation. I wonder how the surface storage capacity is described and parameterized and whether Wsurface and Winteral are completely independent water pools? See also Porada et al. (2018).

Surface storage of Sphagnum is described in Eq. 2 (the ELM default algorithms for representing canopy water, details as described by Oleson et al., 2013). The Sphagnum moss canopy water (canopy_water) is simulated by a function of interception, canopy drip, dew (was added to L240) and canopy evaporation.We treat Wsurface and Winternal as independent pools. Porada et al. (2018) used a process-based numerical simulation model to show that non-vascular vegetation contributes substantially to global rainfall interception and it was an interesting paper.

Third, the authors should describe how evapo(transpi)ration from Sphagnum-PFT is modeled and how it differs from vascular-PFT's. From which water pools evaporation takes place and how evaporation rate or surface conductance depend on Sphagnum characteristics and near-ground microclimate. How and whether evaporation is restricted with decreasing water content? This is required to understand e.g. how SLA and leaf C:N ratio can affect evapotranspiration and interpret the results of sensitivity analysis in Fig 2.

We use the same framework as for vascular PFTs (as described in the new Section 2.2 Non-vascular plants: Sphagnum mosses, L184-199), but the Ball-Berry slope term is assumed to be zero and the intercept term is the conductance term as a function of water content. Drying impacts the conductance and affects evaporation of the internal water. The SLA and leaf C:N ratio parameters are strong controls on Vcmax, and therefore overall productivity and Sphagnum moss LAI. The high sensitivities occur because LAI is a strong control on evapo(transp)iration.

2) Modeling Sphagnum photosynthesis Standard Farquhar-approach is used to simulate Sphagnum net CO2 demand given air chloroplast conductance described as non-linear function of Wtot (eq. 6, from Williams and Flanagan, 1998). In addition, submerging of Sphagnum is assumed to 'kill' CO2 diffusion and thus a restriction to photosynthetic uptake is applied and described as linear function of submerged to total photosynthesizing height (here 0.05m) of the moss. Does the implementation of moss photosynthesis follow Walker et al. (2017)?

Walker et al. also uses the conductance equation from Williams and Flanagan but has a different implementation of the Farquhar model and did not calculate evaporation from the Sphagnum surface. We have added "Submergence in Walker et al. (2017) was expressed as photosynthesising stem area index (SAI) as a logistic function of water table depth. Maximum SAI of 3 was used and the parameter combination that most closely described the GPP data gave a range of water table depth from -10 cm for complete submergence and SAI of ~2.5 at 10 cm. This allowed for a range of processes such as floatation of Sphagnum with the water table, and adhesion of water to the Sphagnum capitula." to main text L308-313.

I like the approach but wonder whether the relatively poor match between modeled and 'measured' moss GPP (Fig 3) can be due i) to ill-represented or omitted temperature response or seasonal acclimation of Vcmax etc., ii) biased Sphagnum temperature (how was it modeled – from surface energy balance?) or ii) too strong submerge-impact. As Sphagnum moss has high leaf (or shoot) area, radiation decays rapidly with canopy depth and thus the top centimeter(s) of the shoot system are responsible of majority of photosynthetic activity. For instance, Niinemets and Tobias (2014) and Zotz and Kahler (2007) show light attenuation profiles and photosynthesis profiles for some moss species. Considering typical characteristics (color) of Sphagnum-canopy, assuming CO2 uptake is evenly distributed across top 5cm may lead to overestimated submerge-impact.

We use the default formulation for acclimation of Vcmax in ELM which is based on a 10-day mean growing temperature. At this point we don't have sufficient measure-
ments to test this assumption,but we can prioritize these measurements in the future. Sphagnum temperature is computed from surface energy balance but because we don't consider the shading effects from trees and shrubs, this may be overestimated. Biases in predicted water table height contribute to errors in the calculated submergence effect. Improving these biases and assuming an exponential rather than a linear $CO_2$ uptake profile may improve representation of the submergence effect. All these aspects may be attribute to the biases of simulated Sphagnum GPP We can consider this in the future when we have more detailed measurements. We have added this content to discussion Section 5.1 L 652-662.

I also wonder whether the soil-respired $CO_2$ leads the Sphagnum to operate in $CO_2$ enriched atmosphere already in current conditions and whether this would lead to overestimated increase of GPP at 900 ppm as photosynthetic $CO_2$-response curve has saturating shape?

"Preliminary isotopic measurements imply a significant fraction of carbon assimilated by the moss may come from subsurface respired $CO_2$ (i.e., $CO_2$ with older 14C signatures predating bomb carbon that can only be sourced from deeper peat, Hanson et al. 2017). However, the observed elevated $CO_2$ response is smaller than simulated (Hanson et al., 2020). Understanding the drivers of elevated $CO_2$ response or lack thereof is a key topic for future work and we will consider this effect in future assessments of the isotopic C budgets for the SPRUCE study." was added to L856-863.

In results L530-534 it is stated that modeled Sphagnum biomass correlates with water table and best correlation is found at with 3-month timelag. For GPP and NPP the instantaneous dependence on WT is from Fig 1. and eq. 6. Please describe how NPP is allocated into biomass and how the growth dynamics of Sphagnum-PFT is modeled; can this explain the timelag?

"NPP is allocated instantaneously into biomass. A positive NPP anomaly caused by water table shifts leads to higher LAI, which also increases future productivity for some

amount of time even if the water table returns to normal. Sphagnum biomass has a 1-year turnover time in the simulation. This combination of effects leads to a roughly 3-month timelag." has been added to L606-610.

3) Modeled carbon cycle components and responses to warming and elevated CO2 For the reader to understand the modeled carbon cycle responses, it is necessary that ELM 'tiling scheme', pathway from NPP to biomass growth and between-PFT competition are better described in Section 2.1 and/or 3.3. That is, present information such as L627-634 earlier in the manuscript. Are shrubs and Sphagnum present as independent tiles or do they occur below the overstory trees?

The default ELM has 16 PFTs and bare ground. For this study, we only included 4 PFTs which are the dominant PFTs for our study site, including boreal evergreen needleleaf tree (Picea), boreal deciduous needleleaf tree (Larix), boreal deciduous shrub (representing several shrub species), and the newly introduced Sphagnum moss PFT (we already mentioned in 3.3 Section, L363-366). Based on the reviewer's suggestion, we moved the related content 'Currently ELM_SPRUCE does not include light competition among multiple PFTs, and thus does not represent cross-PFT shading effects. Our model also allows the canopy density of PFTs to change prognostically, and their fraction cover held constant.' from the original L 627-634 to Section 3.3 L 366-369.

4) Title: "Modeling the hydrology and physiology of Sphagnum moss in a northern temperate bog" should be revised to match the manuscript content. The study is on extending the land-surfacemodel with Sphagnum-PFT and simulating response of moss and vascular vegetation productivity to warming and increasing atmospheric CO2.

We plan to use this as the title "Extending a land-surface model with Sphagnum moss to simulate responses of a northern temperate bog to whole-ecosystem warming and elevated CO2".

Specific comments: L98: water and exchanges within peatland and between peatland and atmosphere?

we already modified the related content to 'water and exchanges within peatland and between peatland and atmosphere (L100-101).'

L 146-147: new chapter – study Aims.

A new paragraph to show the study objective starts with L157 'In this study, we introduce a new Sphagnum moss PFT into the model...' as suggested.

L178-179: Evaporation depends on evaporative demand (VPD; available energy), moss-atmosphere conductance (moss canopy structure, roughness and flow characteristics) and available water pool. The latter is then depends on capillary rise from water table.

We rewrote the related content to 'Since evaporation at the Sphagnum surface depends on atmospheric water vapor deficit, moss-atmosphere conductance and available water pool which depends on capillary wicking of water up to the surface' (L211-213)

L196: canopy_water _ can_water

Thank you for catching this point. We changed canopy_water to can_water (L239).

L211: eq. 6 uses total water content, not Winternal

We used the total water content to calculate the total conductance to $CO_2$ in equation 6, which is consistent with Williams and Flanagan (1998) and Goetz and Price (2015). We reorganized this paragraph and got rid of 'The internal water content of Sphagnum mosses is observed to affect photosynthesis by constraining the length of the diffusive path for $CO_2$ through the variably-hydrated external hyaline cells to the carbon fixation sites (Robroek et al., 2009; Rydin and Jeglum, 2006)'. (L256-264)

L238-239: this assumes boundary-layer conductance » moss surface – chloroplast conductance; assumption is ok but could be mentioned. Note also that maximum g_tc may vary among Sphagnum species?

We added the related content "To be noted that we assume that the boundary layer

conductance is greater than moss surface layer conductance, and the moss surface layer conductance is greater than chloroplast conductance." to the manuscript to L302-304.

L284: what is pre-treatment data?

Pre-treatment data is the data which was collected prior to initiation of the warming and $CO_2$ treatments, and this was added to L349-350.

L363-367: please elaborate whether the data used in parameter optimization is independent of data used in model testing (Fig. 3-4)

The sphagnum GPP in Fig. 3 was not used in the parameter optimization. For the Fig.4, the sphagnum NPP of year 2015-2017 is independent of the optimization, and only above biomass of trees and stem carbon of shrub for year 2012 and 2013 was used for the optimization. We added the years for the data which were used to constrain the model (L433-437), and also added the explanation to Fig.3 and 4 legend.

L393: point should be (*)

Thank you for pointing this out. We changed from point to * (L463).

L479-480: Just curious - why year 2012 was an exception? Were env. drivers different?

Sphagnum production in 2012 was high primarily because of especially high productivity in the hollows during the summer.We double checked the climatical forcing data and did not find the temperature and precipitation were abnormal for year 2012.

L522: Fig. 5: what is driving the strong inter-annual variability of Sphagnum and shrub NPP (annual variability has different sign among these PFT's). Is this mainly due to WT height and does root zone water content affect vascular PFT photosynthesis (O2-stress in wet conditions)?

There are strong inter-annual variabilities of Sphagnum and shrub NPP. For example:the variabilities of Sphagnum and shrub have different signs for years 2020 and

2021 (Fig.5). We compared the BTRAN (representing soil water stress) of shrub for these two years and found that BTRAN may be the driving factor of shrub's variability. The hummock Sphagnum inter-annual variability is mainly driven by water table height with about 3-month lag (Fig.6). The hollow Sphagnum NPP of year 2020 for +0.00oC, +2.25oC, + 4.5oC and +6.75oC temperature levels is lower than the corresponding NPP of year 2021, but it is the opposite way for the +9.00oC condition. The water table is higher for year 2020 than that of year 2021.This implicated that the submerge effect influences the inter-annual variability of hollow Sphagnum NPP. But the inter-annual variabilities are very complicated and it is out of our scope for this study. Thus, we do not plan to include this content to the manuscript text. In addition, we don't currently model the effects of O2 stress in the root zone.

L616-618: this is quite trivial result as Sphagnum water content was made proportional to soil water content (and hence WT).

We changed "Sphagnum growing on hummocks, on the other hand, showed negative warming responses and strong dependency on water table height" to "Sphagnum growing on hummocks, on the other hand, showed negative warming responses and related to the strong dependency on water table height." (L776-778).

L659: The question is that to which extent the parameterization from S1-Bog be generalized to other peatlands?

The algorithms used to represent moss (e.g. Williams and Flanagan) are transferable to and have been applied by other modeling groups in other peatlands. However, we expect that certain parameters will vary, for example, the microtopographic parameters, the relationship between peat moisture and internal water content, and moss properties such as C:N ratio. The parameter sensitivity analysis informs us as to the most important parameters responsible for prediction uncertainties, and can inform how to prioritize these measurements. Collecting these measurements from a variety of sites will be a necessary preliminary exercise (L 916-923).

L667: See e.g. Beringer et al. (2001) and Porada et al. (2016) who have already done this.

Thanks for pointing these two literatures. We added them to the text and listed as references (L738-739).

---

## Author Response (AR1)

*Associate Editor Decision: Reconsider after major revision by Sebastiaan Luyssaert*
*Comments to the Author:*
*Dear authors,*
*Based on the referee comments and the subsequent discussion, I would like to invite you to prepare and upload a revised version of the manuscript. In addition to addressing the main comments of the referees the revision should address the following:*

   A. *Ensure that the knowledge gain in biogeochemical processes is stressed throughout the manuscript.*
   B. *Given the scriticism of both referees on section 5.3 of the discussion, the discussion should be balanced towards biogeochemical processes.*
   C. *Other model approches(OrCHIDEE, JULES,...) should not simply be mentioned in the manuscript but the key differences between those approaches and yours should listed. The readers will want to understand what makes your approach unique.*

Thank you very much for encouraging us to revise the manuscript and good comments.
   A. We first improve our model to include non-vascular plants and then use the updated model to explore the treatment responses, including different warming levels under both elevated and ambient atmospheric CO2 concentration conditions. Our main goal is to explore how the carbon dynamics of the peatland system will change under the treatment conditions. Knowledge about biogeochemical processes is gained by including the moss plant functional type and investigating its impacts on simulated carbon, energy and water cycling in this important peatland ecosystem under conditions approximating future climate.
   B. We agree that the section 5.3 is more focusing on future model development and was detached from the discussion. Followed the comments, we merged section 5.3 to section 5.1 and 5.2 to make the manuscript more coherent with a focus on biogeochemical processes and clearer structure.
   C. Thank you for your good comment about this point. We added those model approaches to the Introduction section (L137-146), and also pointed out how they differ from our model approach.

Reviewer 1

*Even though, mosses are ubiquitous part of boreal vegetation, especially so in peatlands, mosses and their contribution to ecosystem functions are overlooked. The study introduces new plant functional type (PFT) with Sphagnum-specific processes that can be in some extent to be used to describe mosses in other boreal and arctic environments, e.g. upland forests and wet tundra. Manuscript consists, sensitivity analysis of updated land model component and validation part that takes place in boreal ombrotrophic, raised-dome bog peatland with warming and $CO_2$ enrichment experiment. Authors have stated that drier and warmer future climates can lower water table and it has implications on growth of Sphagnum. In the study, capillary rise is a function of peat water content in 10 cm, but it is not clearly stated that if measured or modelled values of peat water content is used in sensitivity analysis and in case study and how water table fluctuations affect functions (e.g. gross primary production) of Sphagnum mosses?*

We use modeled values of peat water content in both the forward simulations and the sensitivity analysis. The modeled water table (WT) has a direct effect on the sphagnum GPP in our model when WT is above the soil surface, through submergence effects (manuscript Eq 7). WT also has an indirect effect on sphagnum GPP, through total conductance to $CO_2$ ($g_{te}$), as follows: $g_{te}$ increases with total Sphagnum water content (manuscript Eq. 6) while GPP increases with $g_{te}$ (manuscript Eqs. 5 and 4). Total sphagnum water content includes a component from Sphagnum internal water (manuscript Eq 3) which is an empirically derived function of soil water nearest the 10cm soil horizon (manuscript Eq 2). As WT drops below the 10cm soil horizon the water content in that layer declines, leading to lower sphagnum internal water, lower sphagnum total water, lower $g_{te}$, and lower GPP.

*If there is more Larix and taller shrubs growing on the site, does it drain more or less the site? Do you see effect of warmer climate on water table depth in higher temperatures and how this will affect Sphaghnum mosses?*

From manuscript Fig.6, we can see the relative biomass changes of Larix, shrub and hollow Sphagnum increase with temperature, the hummock Sphagnum biomass decreases with warming and is more dependent with water table height, and the water table generally decreases with temperature. We have added "We plotted the predicted canopy evaporation for hummock and hollow Sphagnum responses to warming and found that both hummock and hollow Sphagnum canopy evaporation increase with warming for both ambient and elevated atmospheric $CO_2$ conditions despite the Larix and shrubs are growing with warming. Moreover, the hollow Sphagnum canopy evaporation warming response is stronger than that of the hummock Sphagnum (Fig. S2)." to the text (L619-624).

*What I am after is that which kind of hydrological feedbacks there are and how it affects ecophysiology of Sphagnum PFT if temperature will increase +9.0 degrees of Celsius. This is something to think about especially if capillary connection of Sphagnum is described through a simple relationship between capitulum water content and peat water content at 10 cm depth. This could be answered simply by studying hydrological balance of Sphagnum PFT and showing how large part capillary rise plays in Sphagnum hydrological balance. Is it even necessary or which kinds of implications it has to photosynthetic capacity or other ecophysiological processes? In my opinion, authors have not clearly showed or discussed underlying assumptions and consequences of made choices and it should be improved.*

To make how the hydrological cycle affects the Sphagnum ecophysiology clear, we have added "One key feedback is if the water table declines, there can be enhanced decomposition and subsidence of the peat layer, which brings the surface down closer to the water table again. But we currently did not consider the peat layer elevation changes in our model and this will be one of the future development directions. The capillary rise plays into the Sphagnum hydrological balance, which varies depending on water table depth and evaporative demand. At short timescales or under rapidly changing conditions, there may not be equilibration between the Sphagnum water content and the peat moisture. Generally, the Sphagnum water content will equilibrate with the peat on a daily basis outside the plot since the dew point is often reached at

night. But inside the warmer plots since the VPD does not go to zero some disequilibration could remain. High-frequency latent heat flux data from the site are currently lacking, but could help to constrain these effects in the future. The current phenology observations also include if sphagnum hummock and hollow are wet or dry, and we could look at the relationship with soil water content sensors in future work." to the text (L 672-686).

We also added one more reference " Druel et al., 2017" to L701", "Thus, for the Sphagnum mosses desiccation occurs and the time needed before recovery to optimum photosynthetic capacity should be taken into account in our future work" to L 705-707, and "Larmola et al. (2014) also reported that the activity of oxidizing bacteria provides not only carbon but also nitrogen to peat mosses and, thus, contributes to carbon and nitrogen accumulation in peatlands, which store approximately one-third of the global soil carbon pool. We currently didn't consider this kind of CH4 associated carbon and nitrogen uptake by Sphagnum" to L713-720. We will eventually treat the Sphagnum mosses as the "top" soil layer with a lower thermal conductivity and higher hydraulic capacity than a mineral soil layer.

*Sphagnum mosses are sitting on top of high CO2 (and water vapor) sources and experiencing naturally higher concentrations of CO2. How this affects to gross primary production of mosses and which kind of differences there possibly are between mosses that are located to hollows and hummocks? How does this fit to CO2 enrichment study?*

To clarify the elevated $CO_2$ concentration responses of Sphagnum, we add the following text (L828-843): "Sphagnum mosses are sitting on top of high $CO_2$ sources. $CH_4$ can be a significant carbon sources of submerged Sphagnum (Raghoebarsing et al., 2005; Larmola et al, 2014); refixation of $CO_2$ derived from decomposition processes also is an important source of carbon for *Sphagnum* (Rydin and Clymo, 1989; Turetsky and Wieder, 1999). The effects of the elevation of atmospheric $CO_2$ on Sphagnum moss are currently disputed, with studies indicating an increase in growth rate (Jauhiainen and Silvde 1999; Heijmans et al. 2001a; Saarnio et al. 2003), decreases in growth rate (Grosvernier et al. 2001; Fenner et al. 2007) and no response (Van der Hejiden et al. 2000; Hoosbeek et al. 2002; Toet et al. 2006). Norby et al. (2019) indicated that no growth stimulation of both hummock and hollow *Sphagnum* under elevated $CO_2$ condition, but significant negative effects of elevated $CO_2$ on *Sphagnum* NPP in year 2018 at the same study site. Contrasting responses between Sphagnum species are thought to be coupled with the water availability. In contrast, our model results showed that both hummock and hollow *Sphagnum* growths were stimulated by the elevated $CO_2$ concentration, which may be attributed to the fact that we did not consider the light competition between the PFTS (shrub and tree shading effects) and use a fixed cover fraction of Sphagnum."

*Is CO2 concentration profile assumed to be uniform throughout the canopy profile? Does this have effects on results of simulations?*

We added the following text (L844-860): "The $CO_2$ vertical concentration profile is assumed to be uniform in the simulations. In the experiment, the enclosure's regulated additions of pure $CO_2$ are distributed to a manifold that splits the gas into four equal streams feeding each of the four air handling units (Hanson et al., 2017 Fig. 2a), and is injected into the ductwork of each furnace just ahead of each blower and heat exchanger. Horizontal and vertical mixing within each enclosure homogenizes the air volume distributing the $CO_2$ along with the heated air. The horizontal blowers in the enclosures together with external wind eddies ensure vertical mixing.

We do not have routine automated $CO_2$ concentration data below 0.5m. The moss layer may well be experiencing higher concentrations than assumed by the model, but such an impact will be minimized during daylight hours. Preliminary isotopic measurements imply a significant fraction of carbon assimilated by the moss may come from subsurface respired $CO_2$ (i.e., $CO_2$ with older $^{14}C$ signatures predating bomb carbon that can only be sourced from deeper peat, Hanson et al. 2017). We will consider this effect in future assessments of the isotopic C budgets for the SPRUCE study."

*In Chapter 5.3 authors raise important issues and future directions. To me problem is that now it seems to be detached from the model description and discussions. Could this be embedded better in discussions to make the manuscript more coherent and structure clearer?*

Thank you for your good suggestion. We embedded the context of 5.3 Section to Section 5.1 and 5.2 and changed Section 5.2 from "Predicted warming response uncertainties" to "Predicted warming and elevated $CO_2$ concentration response uncertainties".

*L183: Are measurements of Sphagnum water contents from Sphagnum growing on hollows and hummocks? Were there any differences between these microtopograpichal features on water content in moss? Even though, this is clever way to solve capillary rise issue of mosses in simple manner but is this method applicable in both microtopographical positions? My main concern is that does this approach mask the effects of hydrology that is quite important for Sphagnum ecophysiology (main source of water in hollows and hummocks). How about self-cooling (enhanced evaporation) of Sphagnum covered surfaces due to capillary rise? Does the static approach fail especially in sunny days and which kinds of implications it has to Sphagnum ecophysiology?*

The measurements of *Sphagnum* water content during sensor calibrations were primarily on hummock species but included some hollow species. They were not separated during measurements since we needed an integrated measurement for reference against the automated subsurface sensors. We have added this information to the water content dynamics of Sphagnum mosses Section (L219-224). "Currently, we apply the same method for the hummock and hollow Sphagnum water content prediction and can test the model against the measured data when more data are available. Our model still can predict Sphagnum water content differences between these microtopographies as expected, with the water content of hollows greater than that of hummocks. In addition, our model is able to represent the self-cooling effect, although we do not yet have measurements available to validate the model. The relationship of the differences between vegetation temperature (TV) and 2m air temperature (TBOT) (TV-TBOT) and canopy evaporation for both hummock and hollow Sphagnum demonstrated that the differences of TV-TBOT was negative and the canopy evaporation had a negative relationship with TV-TBOT (Fig. S3). " has been added to L724-734.

*L589-L591 This is not only in case with submerged Sphagnum, but it seems that Sphagnum utilizes CH4 as an indirect source of CO2 (e.g. Larmola et al., 2014: DOI: 10.1073/pnas.1314284111)*

We added "Larmola et al. (2014) also reported that the activity of methane oxidizing bacteria provides not only carbon but also nitrogen to peat mosses and, thus, contributes to carbon and nitrogen accumulation in peatlands, which store approximately one-third of the global soil carbon pool." to the manuscript text and cited this literature (L713-719). We also cited it as reference in L829-830.

*L614-618: Can it be that model parameters of hollow and hummock Sphagnum can differ from each other? How this could affect outcome of simulations? I would quess that Sphagnum growing on hummocks are more drought tolerant and resistant than those species growing in hollows. This could be seen i.e. in different slatop -value and, as discussed by authors, in base rate for maintenance respiration.*

To clarify these aspects we added the following text (L 787-795) "We currently used the same parameters for both hummock and hollow, but could consider species differences in the future. Norby et al. (2019) investigated different Sphagnum species at the same site and reported there was no support for the hypothesis that species more adapted to dry conditions (e.g., S. magellanicum and Polytrichum mainly on hummocks) would be more resistant to the stress and would increase in dominance, and both hummock and hollow sphagnum are declining with warming despite the differences between them. This declining trend may be in part due to increased shading from the shrub layer, which is expanding with warming." ELM is currently not able to represent this shading effect and we will address this in future model development.

*L684: Is N fixation somehow represented in a model. Should that be mentioned in a model description? In my opinion, this is quite interesting and important part why moss PFTs should be included in models handling boreal and arctic regions.*

We added "Inputs of new mineral nitrogen of ELM are from atmospheric deposition and biological nitrogen fixation. The fixation of new reactive nitrogen from atmospheric N2 by soil microorganisms is an important component of nitrogen budgets. ELM follows the approach of Cleveland et al. (1999) that uses an empirical relationship of biological nitrogen fixation as a function of net primary production to predict the nitrogen fixation" to Section 2.1(L177-183). We also added "We are measuring Sphagnum associated N2 fixation at the SPRUCE site and found that rates decline with increasing temperature (Carrell et al. 2019 Global Change Biology). We are continuing these measurements to see if they correlate with the GPP empirical relationship from Clevand (1999), or if temperature disrupts that association. Once finished, results will be used to represent N fixation by the Sphagnum layer and testing with measurements." to L870-873 and L890-891.

Reviewer 2

*In this study, new plant functional type (PFT) describing Sphagnum –moss, abundant in boreal and arctic peatlands, is incorporated into the land model component ELM of the Earth System model E3SM to better represent carbon, water and nutrient cycling in boreal and arctic regions.*

*The ELM with the newly proposed Sphagnum-PFT was parameterized and evaluated against data collected under ambient conditions and under climate-change experiment conducted at an ombrothrophic bog in Minnesota, US. Further, the model is used to predict changes in moss and vascular plant productivity, biomass accumulation and water table level for combined temperature and CO2 increase scenarios.*

*The article is well written and fits topically under the scope of BG. However, the relevance for larger scientific community is limited as the study is centered on development of a specific land model and testing for a specific site. I therefore consider the study as border line case for BG and maybe fits better into a more specific model development journal such as Geoscientific Model Development. However, this is up to the Editor to decide.*

The primary goal of the SPRUCE project is to test how vulnerable an important Carbon-rich terrestrial ecosystem is to atmospheric and climatic change by warming the entire soil profile and measuring whether large amounts of $CO_2$ and $CH_4$ are emitted   The regression design allows the derivation of  key temperature response functions for mechanistic ecosystem processes that can be used for model validation and improvement.  In this study, we introduce a moss PFT into the land model component (ELM) of the Energy Exascale Earth System Model (E3SM). Then, we evaluate our updated model against numerous measurements. We also apply the updated ELM to explore how an ombrotrophic, raised-dome bog peatland ecosystem will respond to different scenarios of warming and elevated atmospheric $CO_2$ concentration. The model development is only part of our goal, and we mainly focus on using the model to investigate the peatland ecosystem responses to changing climate and the feedbacks.  Our model development work at SPRUCE in this manuscript is a first step to a broader peatland model in E3SM that can predict key climate feedbacks from these important ecosystems.  The broader representativeness of the ecosystem responses at SPRUCE for other peatland systems was considered in the design of the experiment and will be further assessed using ELM-SPRUCE in future work at additional sites.

*In recent years there has been strong interest on including Sphagnum as well as feather mosses and other bryophytes into land-surface models. In addition to the references listed in the Introduction the authors should take a look and cite the recent works of Philip Porada and colleagues (Porada et al., 2013, 2016), as well as note the inclusion of moss-PFT into ORCHIDEE-model (Druel et al., 2017).  The authors should also be more explicit how their study builds on and improves the existing knowledge and methods to describe Sphagnum mosses in land surface models. If the study is to be published in BG, the results and methods should in my opinion be generalized and better interpreted against existing literature. Currently, the discussion, in particular Section 5.3, reads more as a research plan for future development of a specific land surface model.*

Thank you for the suggestion to place our work in the context of existing literature.  We added the following text and citations to the introduction (L137-146): "Druel et al. (2017) investigated the vegetation-climate feedbacks in high latitudes by implementing the nonvascular plants including bryophytes and lichens to the global land surface model ORCHIFEE.  Porada et al. (2016) integrated a stand-alone dynamic non-vascular vegetation model LiBry (Porada et al.,

2013) to land surface scheme JSBACH, but LiBry and JSBACH mainly represent bryophyte and lichen growth on upland forest floor sites, not for wetland sites. Chadburn et al. (2015) introduced a new moss PFT to JULES land surface model and treated the thermal conductivity of moss depending on its water content
We also added a new section '2.2 Non-vascular plants: Sphagnum mosses' to Section 2 Model description to describe more details how we implement our Sphagnum mosses into our model (L189-202). For the future model development Section 5.3, we embedded it into Section 5.1 and 5.2 as the other reviewer also suggested.

*My general comments are as follows:*
*1) Modeling Sphagnum water content*
*Sphagnum total water content is sum of two pools: Wtot = Winternal + Wsurface There are few remarks / comments that should be made. First, Winternal is described as non-linear function of top soil water content and thus immediately adjust to changes in soil water content (or water table). This approach thus assumes that in Sphagnum, Winternal is at hydrostatic equilibrium with soil water potential (or water content) as defined through water-retention characteristics of the peat-Sphagnum continuum. Moreover, it assumes that hydraulic conductivity is sufficiently large so that Sphagnum water content is never decoupled from soil water content. Such assumptions may not hold in case water table (WT) drops deep during prolonged dry periods, more propable in future climat*

We added the following text (L686-691)"Moreover, the equilibration time between peat moisture and moss water content is reasonably fast, but the timescales for rewetting should change as the peat dries since the cross section for capillary rise will decline and thus the maximum flux to the surface will decline. So, at some point, between gravity potential and reduced hydraulic conductivity the capillarity will not satisfy evaporative demand."
 But for the simplicity, we currently used the empirical representation of water content in our model for both hummock and hollow Sphagnum (as we responded above for the first viewer's comment).
We also added "Moreover, Walker et al., (2017) reported that the function of Sphagnum water content to soil water content or to water table depth they used for the SPRUCE site was empirical and may not be representative for peatland ecosystem.  To better represent the peatland ecosystem in our model, we will eventually treat the Sphagnum mosses as the "top" soil layer with a lower thermal conductivity and higher hydraulic capacity." L734-739.

*What is author's conclusion on generality of Winternal – soil water content relationship (Fig. 1) among Sphagnum species (hummock vs. hollow –preferences)? And how Winternal and Wexternal pools were separated from the gravimetric measurements of water content in Sphagnum to derive relationship between Winternal and soil water content?*

During the calibrations, we used intact monoliths collected from multiple locations. The monoliths included both hummock and hollow species, but they were not separated during destructive measurements, since we needed an integrated measurement for comparison against the subsurface soil water content sensors. We have clarified this information to the water content dynamics of Sphagnum mosses Section (L219-224).

*Second, the Wsurface is filled by interception of rainfall (how about condensation?) and drained by evaporation. I wonder how the surface storage capacity is described and parameterized and whether Wsurface and Winternal are completely independent water pools? See also Porada et al. (2018).*

Surface storage of Sphagnum is described in Eq. 2 (the ELM default algorithms for representing canopy water, details as described by Oleson et al., 2013). The Sphagnum moss canopy water (canopy_water) is simulated by a function of interception, canopy drip, dew (was added to L244) and canopy evaporation.We treat Wsurface and Winternal as independent pools. Porada et al. (2018) used a process-based numerical simulation model to show that non-vascular vegetation contributes substantially to global rainfall interception and it was an interesting paper.

*Third, the authors should describe how evapo(transpi)ration from Sphagnum-PFT is modeled and how it differs from vascular-PFT's. From which water pools evaporation takes place and how evaporation rate or surface conductance depend on Sphagnum characteristics and near-ground microclimate. How and whether evaporation is restricted with decreasing water content? This is required to understand e.g. how SLA and leaf C:N ratio can affect evapotranspiration and interpret the results of sensitivity analysis in Fig 2.*

We use the same framework as for vascular PFTs (as described in the new Section 2.2 Non-vascular plants: Sphagnum mosses, L189-202), but the Ball-Berry slope term is assumed to be zero and the intercept term is the conductance term as a function of water content.   Drying impacts the conductance and affects evaporation of the internal water.  The SLA and leaf C:N ratio parameters are strong controls on Vcmax, and therefore overall productivity and Sphagnum moss LAI.  The high sensitivities occur because LAI is a strong control on evapo(transp)iration.

*2) Modeling Sphagnum photosynthesis*
*Standard Farquhar-approach is used to simulate Sphagnum net CO2 demand given air chloroplast conductance described as non-linear function of Wtot (eq. 6, from Williams and Flanagan, 1998). In addition, submerging of Sphagnum is assumed to 'kill' CO2 diffusion and thus a restriction to photosynthetic uptake is applied and described as linear function of submerged to total photosynthesizing height (here 0.05m) of the moss. Does the implementation of moss photosynthesis follow Walker et al. (2017)?*

Walker et al. also uses the conductance equation from Williams and Flanagan but has a different implementation of the Farquhar model and did not calculate evaporation from the Sphagnum surface.  We have added "Submergence in Walker et al. (2017) was expressed as photosynthesising stem area index (SAI) as a logistic function of water table depth. A maximum SAI of 3 was used and the parameter combination that most closely described the GPP data gave a range of water table depth from -10 cm for complete submergence and SAI of ~2.5 at 10 cm. This allowed for a range of processes such as floatation of Sphagnum with the water table, and adhesion of water to the Sphagnum capitula." to main text L312-317.

*I like the approach but wonder whether the relatively poor match between modeled and 'measured' moss GPP (Fig 3) can be due i) to ill-represented or omitted temperature response*

*or seasonal acclimation of Vcmax etc., ii) biased Sphagnum temperature (how was it modeled – from surface energy balance?) or ii) too strong submerge-impact. As Sphagnum moss has high leaf (or shoot) area, radiation decays rapidly with canopy depth and thus the top centimeter(s) of the shoot system are responsible of majority of photosynthetic activity. For instance, Niinemets and Tobias (2014) and Zotz and Kahler (2007) show light attenuation profiles and photosynthesis profiles for some moss species. Considering typical characteristics (color) of Sphagnum-canopy, assuming CO2 uptake is evenly distributed across top 5cm may lead to overestimated submerge-impact.*

We added the following text to discussion Section 5.1 L 656-666 "In addition, we use the default formulation for acclimation of Vcmax in ELM which is based on a 10-day mean growing temperature.  At this point we don't have sufficient measurements to test this assumption,but we can prioritize these measurements in the future.  Sphagnum temperature is computed from surface energy balance but because the current model doesn't have the capacity to estimate shading effects from trees and shrubs, this may be overestimated.  Moreover, biases in predicted water table height contribute to errors in the calculated submergence effect.  Improving these biases and assuming an exponential rather than a linear $CO_2$ uptake profile may improve representation of the submergence effect.  All these aspects may be attribute to the biases of simulated Sphagnum GPP. We can consider this in the future when we have more detailed measurements."

*I also wonder whether the soil-respired CO2 leads the Sphagnum to operate in CO2 enriched atmosphere already in current conditions and whether this would lead to over-estimated increase of GPP at 900 ppm as photosynthetic CO2-response curve has saturating shape?*

"Preliminary isotopic measurements imply a significant fraction of carbon assimilated by the moss may come from subsurface respired $CO_2$ (i.e., $CO_2$ with older $^{14}C$ signatures predating bomb carbon that can only be sourced from deeper peat, Hanson et al. 2017). However, the observed elevated $CO_2$ response is smaller than simulated (Hanson et al., 2020). Understanding the drivers of elevated $CO_2$ response or lack thereof is a key topic for future work and we will consider this effect in future assessments of the isotopic C budgets for the SPRUCE study." was added to L854-860.

*In results L530-534 it is stated that modeled Sphagnum biomass correlates with water table and best correlation is found at with 3-month timelag. For GPP and NPP the instantaneous dependence on WT is from Fig 1. and eq. 6. Please describe how NPP is allocated into biomass and how the growth dynamics of Sphagnum-PFT is modeled; can this explain the timelag?*

"NPP is allocated instantaneously into biomass.  A positive NPP anomaly caused by water table shifts leads to higher LAI, which also increases future productivity for some amount of time even if the water table returns to normal.  Sphagnum biomass has a 1-year turnover time in the simulation.  This combination of effects leads to a roughly 3-month timelag." has been added to L610-614.

*3) Modeled carbon cycle components and responses to warming and elevated CO2*
*For the reader to understand the modeled carbon cycle responses, it is necessary that ELM 'tiling scheme', pathway from NPP to biomass growth and between-PFT competition are better described in Section 2.1 and/or 3.3. That is, present information such as L627-634 earlier in the manuscript. Are shrubs and Sphagnum present as independent tiles or do they occur below the overstory trees?*

The default ELM has 16 PFTs and bare ground. For this study, we only included 4 PFTs which are the dominant PFTs for our study site, including boreal evergreen needleleaf tree (Picea), boreal deciduous needleleaf tree (Larix), boreal deciduous shrub (representing several shrub species), and the newly introduced Sphagnum moss PFT (we already mentioned in 3.3 Section, L367-370). Based on the reviewer's suggestion, we moved the related content 'Currently ELM_SPRUCE does not include light competition among multiple PFTs, and thus does not represent cross-PFT shading effects. Our model also allows the canopy density of PFTs to change prognostically, and their fractional coverage is held constant.' from the original L 627-634 to Section 3.3 L 370-373.

*4) Title: "Modeling the hydrology and physiology of Sphagnum moss in a northern temperate bog" should be revised to match the manuscript content. The study is on extending the land-surfacemodel with Sphagnum-PFT and simulating response of moss and vascular vegetation productivity to warming and increasing atmospheric CO2.*

We plan to use this as the title "Extending a land-surface model with Sphagnum moss to simulate responses of a northern temperate bog to whole-ecosystem warming and elevated $CO_2$".

*Specific comments:*
*L98: water and exchanges within peatland and between peatland and atmosphere?*

we already modified the related content to 'water and exchanges within peatland and between peatland and atmosphere (L100-101).'

*L 146-147: new chapter – study Aims.*

A new paragraph to show the study objective starts with L160 'In this study, we introduce a new Sphagnum moss PFT into the model…' as suggested.

*L178-179: Evaporation depends on evaporative demand (VPD; available energy), moss-atmosphere conductance (moss canopy structure, roughness and flow characteristics) and available water pool.*
*The latter is then depends on capillary rise from water table.*

We rewrote the related content to 'Since evaporation at the Sphagnum surface depends on atmospheric water vapor deficit, moss-atmosphere conductance and available water pool which depends on capillary wicking of water up to the surface' (L215-217)

*L196: canopy_water _ can_water*

Thank you for catching this point. We changed canopy_water to can_water (L243).

*L211: eq. 6 uses total water content, not Winternal*

Yes, we used the total water content to calculate the total conductance to $CO_2$ in equation 6, which is consistent with Williams and Flanagan (1998) and Goetz and Price (2015). Thus, we removed 'The internal water content of Sphagnum mosses is observed to affect photosynthesis by constraining the length of the diffusive path for $CO_2$ through the variably-hydrated external hyaline cells to the carbon fixation sites (Robroek et al., 2009; Rydin and Jeglum, 2006)'.

*L238-239: this assumes boundary-layer conductance >> moss surface – chloroplast conductance;*
*assumption is ok but could be mentioned. Note also that maximum g_tc may vary among Sphagnum species?*

We added the related content "To be noted that we assume that the boundary layer conductance is greater than moss surface layer conductance, and the moss surface layer conductance is greater than chloroplast conductance." to the manuscript to L306-308.

*L284: what is pre-treatment data?*

Pre-treatment data is the data which was collected prior to initiation of the warming and $CO_2$ treatments, and this was added to L353-354.

*L363-367: please elaborate whether the data used in parameter optimization is independent of data used in model testing (Fig. 3-4)*

The sphagnum GPP in Fig. 3 was not used in the parameter optimization. For the Fig.4, the sphagnum NPP of year 2015-2017 is independent of the optimization, and only above biomass of trees and stem carbon of shrub for year 2012 and 2013 was used for the optimization. We added the years for the data which were used to constrain the model (L437-441), and also added the explanation to Fig.3 and 4 legend.

*L393: point should be (\*)*

Thank you for pointing this out. We changed from point to * (L467).

*L479-480: Just curious - why year 2012 was an exception? Were env. drivers different?*

Sphagnum production in 2012 was high primarily because of especially high productivity in the hollows during the summer.We double checked the climatical forcing data and did not find the temperature and precipitation were abnormal for year 2012.

*L522: Fig. 5: what is driving the strong inter-annual variability of Sphagnum and shrub NPP (annual variability has different sign among these PFT's). Is this mainly due to WT height and does root zone water content affect vascular PFT photosynthesis (O2-stress in wet conditions)?*

There are strong inter-annual variabilities of Sphagnum and shrub NPP. For example,the variabilities of Sphagnum and shrub have different signs for predicted years 2020 and 2021 (Fig.5). We compared the BTRAN (a scalar representing soil water stress) of shrub for these two years and found that BTRAN may be the driving factor of shrub's variability. The hummock Sphagnum inter-annual variability is mainly driven by water table height with about 3-month lag (Fig.6). The hollow Sphagnum NPP of year 2020 for +0.00°C, +2.25°C, + 4.5°C and +6.75°C temperature levels is lower than the corresponding NPP of year 2021, but it is the opposite way for the +9.00°C condition. The water table is higher for year 2020 than that of year 2021.This implicated that the submerge effect influences the inter-annual variability of hollow Sphagnum NPP. But many complex factors drive the inter-annual variability, and it is out of our scope for this study. Thus, we do not plan to include this content to the manuscript text. In addition, we don't currently model the effects of $O_2$ stress in the root zone.

*L616-618: this is quite trivial result as Sphagnum water content was made proportional to soil water content (and hence WT).*

We changed "Sphagnum growing on hummocks, on the other hand, showed negative warming responses and strong dependency on water table height" to "Sphagnum growing on hummocks, on the other hand, showed negative warming responses that are related to the strong dependency on water table height." (L776-778).

*L659: The question is that to which extent the parameterization from S1-Bog be generalized to other peatlands?*

The algorithms used to represent moss (e.g. Williams and Flanagan) are transferable to and have been applied by other modeling groups in other peatlands.  However, we expect that certain parameters will vary, for example, the microtopographic parameters, the relationship between peat moisture and internal water content, and moss properties such as C:N ratio.  The parameter sensitivity analysis informs us as to the most important parameters responsible for prediction uncertainty, and can inform how to prioritize these measurements.  Collecting these measurements from a variety of sites will be a necessary preliminary exercise (L 913-920).

*L667: See e.g. Beringer et al. (2001) and Porada et al. (2016) who have already done this.*

Thanks for pointing these two literatures. We added them to the text and listed as references (L739).

---

## Author Response (AR2)

Associate Editor comments:

Comments to the Author:

In the light of the positive assessment of the revised manuscript by the referee, I'm happy to offer publication of your manuscript in Biogeosciences. Note that this offer is conditional on revising the minor issues noted by the referee. Also, I agree with the referee that especially the discussion should be edited to further enhance its readability. Looking forward to the revised manuscript,

**Thank you for the encouraging comments and giving us an opportunity to revise our manuscript. We addressed all the comments and make some edits and corrections as well.**

Referee #2

The authors have made thorough work in revising the MS and well addressed my comments. Thanks for scientific discussion!

**Thank you for your all nice comments and We are happy we addressed all your comments.**

I suggest the MS can be published after proof-reading. I encourage the authors to check whether part of their additions (in red) into Discussion could be shortened.

**Thank you for your suggestion! We double checked the Discussion part, made some changes and got rid of one paragraph to make it more readable.**

Few remarks: L145: ORCHIDEE, L615: capture processes, L913: citation missing!

**We changed 'ORCHIFEE' to 'ORCHIDEE' (L141) and added the citation of Williams and Flanagan, 1998 (L877). But for L 615 'capture processes', we double check the last version of manuscript and didn't capture what kind of response we should give**

Extending a land-surface model with *Sphagnum* moss to simulate responses of a northern temperate bog to whole-ecosystem warming and elevated $CO_2$

Xiaoying Shi[1][*], Daniel M. Ricciuto[1], Peter E. Thornton[1], Xiaofeng Xu[2], Fengming Yuan[1],

Richard J. Norby[1], Anthony P. Walker[1], Jeffrey Warren[1], Jiafu Mao[1], Paul J. Hanson[1],

Lin Meng[3], David Weston[1], Natalie A. Griffiths[1]

[1] Climate Change Science Institute and Environmental Sciences Division, Oak Ridge

National Laboratory, Oak Ridge, TN 37831, USA

[2]Biology Department San Diego State University, San Diego, CA, 92182-4614, USA

[3] Department of Geological and Atmospheric Sciences, Iowa State University, Ames, IA,

50011

[*] To whom correspondence should be addressed

Corresponding author's email: shix@ornl.gov

Fax: 865-574-2232

**Abstract**

Mosses need to be incorporated into Earth system models to better simulate peatland functional dynamics under changing environment. *Sphagnum* mosses are strong determinants of nutrient, carbon and water cycling in peatland ecosystems. However, most land surface models do not include *Sphagnum* or other mosses as represented plant functional types (PFTs), thereby limiting predictive assessment of peatland responses to environmental change. In this study, we introduce a moss PFT into the land model component (ELM) of the Energy Exascale Earth System Model (E3SM), by developing water content dynamics and non-vascular photosynthetic processes for moss. The model was parameterized and independently evaluated against observations from an ombrotrophic forested bog as part of the Spruce and Peatland Responses Under Changing Environments (SPRUCE) project. Inclusion of a *Sphagnum* PFT with some *Sphagnum* specific processes in ELM allows it to capture the observed seasonal dynamics of *Sphagnum* gross primary production (GPP), albeit with an underestimate of peak GPP. The model simulated a reasonable annual net primary production (NPP) for moss but with less interannual variation than observed, and reproduced above ground biomass for tree PFTs and stem biomass for shrubs. Different species showed highly variable warming responses under both ambient and elevated atmospheric $CO_2$ concentrations, and elevated $CO_2$ altered the warming response direction for the peatland ecosystem. Microtopography is critical: *Sphagnum* mosses on hummocks and hollows were simulated to show opposite warming responses (NPP decreasing with warming on hummocks, but increasing in hollows), and hummock *Sphagnum* was modeled to have strong dependence on water table height. Inclusion of this new moss PFT in global ELM

simulations may provide a useful foundation for the investigation of northern peatland carbon exchange, enhancing the predictive capacity of carbon dynamics across the regional and global scales.

**Copyright statement**

This manuscript has been authored by UT-Battelle, LLC under Contract No. DE-

AC05-00OR22725 with the U.S. Department of Energy. The United States Government retains and the publisher, by accepting the article for publication, acknowledges that the

United States Government retains a non-exclusive, paid-up, irrevocable, world-wide license to publish or reproduce the published form of this manuscript, or allow others to do so, for United States Government purposes. The Department of Energy will provide public access to these results of federally sponsored research in accordance with the DOE

Public Access Plan (http://energy.gov/downloads/doe-public-access-plan).

[revised manuscript text omitted]